# Hepatocyte-specific IL11 cis-signaling drives lipotoxicity and underlies the transition from NAFLD to NASH

Jinrui Dong[1], Sivakumar Viswanathan[1], Eleonora Adami[1], Brijesh K. Singh[1], Sonia P. Chothani [1], Benjamin Ng[1,2], Wei Wen Lim [2], Jin Zhou[1], Madhulika Tripathi[1], Nicole S. J. Ko[1], Shamini G. Shekeran[1], Jessie Tan[1,2], Sze Yun Lim[2], Mao Wang [1], Pei Min Lio[2], Paul M. Yen[1], Sebastian Schafer [1,2], Stuart A. Cook [1,2,3,4,5✉] & Anissa A. Widjaja [1,5✉]

IL11 is important for fibrosis in non-alcoholic steatohepatitis (NASH) but its role beyond the stroma in liver disease is unclear. Here, we investigate the role of IL11 in hepatocyte lipotoxicity. Hepatocytes highly express IL11RA and secrete IL11 in response to lipid loading. Autocrine IL11 activity causes hepatocyte death through NOX4-derived ROS, activation of ERK, JNK and caspase-3, impaired mitochondrial function and reduced fatty acid oxidation. Paracrine IL11 activity stimulates hepatic stellate cells and causes fibrosis. In mouse models of NASH, hepatocyte-specific deletion of *Il11ra1* protects against liver steatosis, fibrosis and inflammation while reducing serum glucose, cholesterol and triglyceride levels and limiting obesity. In mice deleted for *Il11ra1*, restoration of IL11 cis-signaling in hepatocytes reconstitutes steatosis and inflammation but not fibrosis. We found no evidence for the existence of IL6 or IL11 trans-signaling in hepatocytes or NASH. These data show that IL11 modulates hepatocyte metabolism and suggests a mechanism for NAFLD to NASH transition.

---

[1] Cardiovascular and Metabolic Disorders Program, Duke-National University of Singapore Medical School, Singapore, Singapore. [2] National Heart Research Institute Singapore, National Heart Centre Singapore, Singapore, Singapore. [3] National Heart and Lung Institute, Imperial College London, London, UK. [4] MRC-London Institute of Medical Sciences, Hammersmith Hospital Campus, London, UK. [5] These authors jointly supervised this work: Stuart A. Cook, Anissa A. Widjaja. ✉email: stuart.cook@duke-nus.edu.sg; anissa.widjaja@duke-nus.edu.sg

nterleukin 11 (IL11) is a fibrogenic factor[1–4] that is elevated in fibrotic precision-cut liver slices across species[5]. IL11 has recently been shown to have negative effects on hepatocyte function after toxic liver insult[6] and, directly or indirectly, contributes to nonalcoholic steatohepatitis (NASH) pathologies[7]. At the other end of the spectrum, a number of earlier publications suggest that IL11 is protective in mouse models of ischemic-, infective-, or toxin-induced liver damage[8–13]. However, it is now apparent that the recombinant human IL11 (rhIL11) reagent used in these earlier studies has unexpected effects in the mouse[6] and the question as to the true biological effect of IL11 in the liver, specifically in hepatocytes, remains open.

IL11 is a member of the interleukin 6 (IL6) cytokine family and binds to its cognate alpha receptor (IL11RA) and then to glycoprotein 130 (gp130) to signal in *cis*. IL6 itself has been linked to liver function and publications suggest an overall beneficial effect[14–19]. Aside from *cis*-signaling, IL6 can also bind to soluble IL6 receptor (sIL6R) and signal in *trans*. IL6 *trans*-signaling is considered maladaptive in the context of metabolic and autoimmune disease but, somewhat paradoxically, beneficial for liver regeneration[16]. It is possible that IL11, like IL6, also signals in *trans* but experiments to date have found no evidence for this in tumors or reproductive tissues[20,21].

The factors underlying the transition from nonalcoholic fatty liver disease (NAFLD) to NASH are multifactorial but lipid loading of hepatocytes is of central importance[22]. Certain lipid species are toxic for hepatocytes and lipotoxicity leads to cytokine release causing hepatocyte death along with activation of hepatic stellate cells (HSCs) and immune cells[22,23]. Lipotoxicity, such as that due to palmitate[24], is an early event in NASH and represents a linkage between diet, NAFLD, and NASH. While genetic or pharmacological inhibition of IL6 *cis*-signaling worsens steatosis phenotypes[17,18,25], a role for IL11 in hepatic lipotoxicity has not been described.

In the current study, we used a range of in vitro and in vivo approaches to address key questions regarding IL11 in hepatocyte biology, NAFLD, and NASH: (1) Defining the role of IL11 signaling in human hepatocytes, (2) examining whether lipotoxicity is related to IL11 activity in hepatocytes, (3) establishing whether IL11 (or IL6) *trans*-signaling contributes to NASH, (4) dissecting the interrelationship between IL11 *cis*-signaling in hepatocytes and the development of steatosis, hepatocyte death, inflammation, and fibrosis. These studies demonstrate a detrimental effect of lipotoxicity-associated IL11 signaling in hepatocytes that appears to be apical pathology in the aetiology of NASH.

## Results

### High levels of IL11RA expression in primary human hepatocytes

We first assessed the expression levels of IL6R and IL11RA in healthy human or mouse liver by immunohistochemistry. In both human and mouse liver sections there was limited staining of IL6R but robust expression of IL11RA, which appeared mostly localized to hepatocytes (Fig. 1a; Supplementary Fig. 2a). This is consistent with staining data from the human protein atlas using two additional antibodies (CAB032830 and HPA036652; https://www.proteinatlas.org). Interestingly, as compared to control livers, IL11RA expression was increased in liver biopsies from patients with NASH and also in livers from mice with NASH on a Western Diet supplemented with fructose (Supplementary Fig. 2b and c).

Flow cytometry studies confirmed that both IL11RA and gp130 are highly expressed in primary human hepatocytes whereas IL6R was expressed at much lower levels (Fig. 1b; Supplementary Fig. 3a). In contrast, IL6R was highly expressed on immune cells where IL11RA expression was low that is in keeping with

previous data showing reciprocal expression of these two receptors on different cells (Fig. 1b)[1,19]. RNA-seq and Ribo-seq studies confirmed *IL11RA* and *gp130* transcripts to be highly expressed and actively translated in hepatocytes, whereas *IL6R* transcripts were low and there was little eveidence of *IL6R* translation (Fig. 1c–e; Supplementary Fig. 3b and c). Immunofluorescence staining of primary human and mouse hepatocytes and some of the most commonly used hepatocyte-like cell lines (HepG2 and AML12) revealed that all of these cells consistently had high IL11RA but low IL6R expression (Supplementary Fig. 3d). Overall, these data show strong co-expression of both IL11RA and gp130 in hepatocytes across species.

### IL11 *cis*-signaling causes hepatocyte death

In hepatocytes, IL11 activated non-canonical signaling pathways (ERK and JNK) in a dose-dependent manner (2.5–20 ng/ml), while IL6 activated STAT3 (Supplementary Fig. 4a). To compare IL6 and IL11 signaling, while circumventing potential complexities associated with different levels of IL11RA or IL6R expression, we used a synthetic IL6 *trans*-signaling construct (hyperIL6) and compared this with a synthetic IL11 *trans*-signaling complex (hyperIL11). HyperIL11, like IL11 itself, dose-dependently activated ERK and JNK. Similarly, IL6 *trans*-signaling dose-dependently induced STAT3 phosphorylation, as seen with IL6 itself, but did not activate ERK or JNK (Fig. 1f). Thus, IL11 or IL6 (*cis* and *trans*) signaling results in activation of different intracellular pathways in hepatocytes.

HyperIL11, like IL11[6], caused a dose-dependent increase in alanine transaminase (ALT) in the media of primary human hepatocyte cell cultures whereas hyperIL6 (20 ng/ml) had a significant, albeit limited, protective effect (ALT fold change (FC) = 0.9; $P = 0.0468$) (Fig. 1g). Soluble gp130 (sgp130) is a selective inhibitor of *trans*-signaling complexes acting through gp130[16]. Consistent with its reported decoy effects, sgp130 blocked the activation of signaling pathways downstream of both hyperIL11 (ERK/ JNK) and hyperIL6 (STAT3) as well as inhibited the hepatotoxic effects of hyperIL11 (Fig. 1h–j).

We next probed for the existence of IL11 *trans*-signaling in a physiological context, in the absence of preformed, synthetic/alien protein complexes. We stimulated cells with IL11 in the presence of either soluble gp130 (sgp130, to inhibit putative *trans*-signaling) or soluble IL11RA (sIL11RA, to potentiate putative *trans*-signaling). IL11-induced caspase-3 activation, NOX4 upregulation, ERK and JNK signaling, and hepatocyte cell death were unaffected by sgp130 or sIL11RA (1 μg/ml) (Fig. 1k and l; Supplementary Fig. 4b). Furthermore, IL11 dose-dependently caused hepatocyte cell death, which was unaffected by the addition of sgp130 or sIL11RA (Supplementary Fig. 4c). Reciprocally, increasing doses of sgp130 or sIL11RA had no effect on ALT release from IL11-stimulated hepatocytes (Supplementary Fig. 4d). These data argue against the existence of IL11 *trans*-signaling in hepatocytes.

Reactive oxygen species and caspases are implicated, together or alone, in lipotoxic cell death[26]. To probe the mechanisms underlying IL11-regulated hepatocyte death we inhibited NOX4 with GKT-13781 or DPI and caspases with Z-VAD-FMK. NOX4 inhibitors reduced IL11-induced ERK and JNK activation and robustly protected hepatocytes from IL11-induced cell death (Supplementary Fig. 4e and f). Pan-caspase inhibition, while protective, was not as effective as NOX4 inhibition in preventing cell death and did not reduce either NOX4 induction or ERK activation (Supplementary Fig. 4g and h). This places NOX4 activity upstream of late-stage (24 h) ERK and caspase-3 activation in IL11-stimulated hepatocytes and suggests that apoptotic cell death is only one mode of cell death in this context.

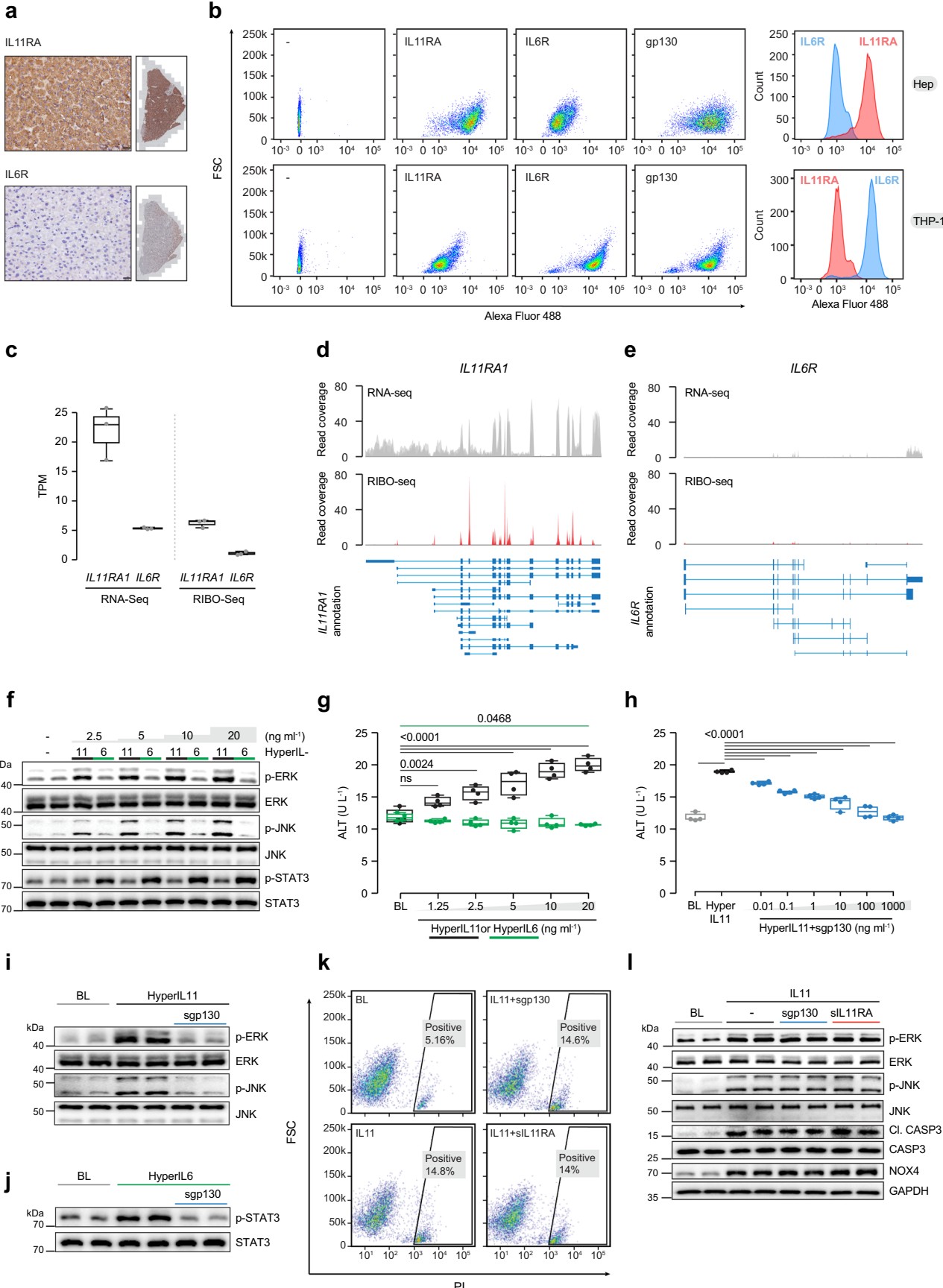

**Fig. 1 IL11RA is highly expressed in hepatocytes and IL11 *cis*-signaling is hepatotoxic. a** Immunohistochemistry staining of IL11RA and IL6R in healthy human liver sections (scale bars, 20 μm, *n* = 1 independent experiment, due to limited amount of human liver section). **b** Flow cytometry forward scatter (FSC) plots of IL11RA, IL6R, and gp130 staining and fluorescence intensity plots of IL11RA and IL6R staining on hepatocytes and THP-1. **c** Abundance of *IL11RA1* and *IL6R* reads in hepatocytes at baseline based on RNA-seq (left) and Ribo-seq (right) (transcripts per million, TPM) (*n* = 3). **d, e** Read coverage of **d** *IL11RA1* and **e** *IL6R* transcripts based on RNA-seq (gray) and Ribo-seq (red) of primary human hepatocytes (*n* = 3). **f** Western blots showing ERK, JNK, and STAT3 activation status and **g** ALT secretion (*n* = 4) by hepatocytes following a dose range stimulation of either hyperIL11 or hyperIL6. **h** ALT levels in the supernatants of hepatocytes stimulated with hyperIL11 alone or in the presence of increasing amounts of soluble gp130 (sgp130) (*n* = 4). **i, j** Western blots of hepatocyte lysates showing **i** phosphorylated ERK and JNK and their respective total expression in response to hyperIL11 stimulation alone or with sgp130 and **j** phospho-STAT3 and total STAT3 in response to hyperIL6 stimulation with and without sgp130. **k** Representative FSC plots of propidium Iodide (PI) staining of IL11-stimulated hepatocytes in the presence of sgp130 or soluble IL11RA (sIL11RA). **l** Western blots showing phospho-ERK, phospho-JNK, cleaved caspase-3, and their respective total expression, NOX4, and GAPDH in hepatocytes in response to IL11 stimulation alone or in the presence of sgp130 or sIL11RA. **i, j, l** Representative data of *n* = 2 independent experiments. **b–l** Primary human hepatocytes; **f–l** 24 h stimulation; hyperIL11, hyperIL6, IL11 (20 ng/ml), sgp130, sIL11RA (1 μg/ml). **c, g, h** Data are shown as box-and-whisker with median (middle line), 25th–75th percentiles (box), and min–max values (whiskers). **g, h** One-way ANOVA with Dunnett's correction. Source data are provided as a Source data file.

**IL11 *cis*-signaling underlies lipotoxicity in hepatocytes.** To begin to examine the role of IL11 in fatty liver disease, we modeled hepatocyte lipotoxicity, widely viewed as an initiating pathology for NASH and related to cytokine release[22]. To do so, we loaded hepatocytes with palmitate using a concentration of saturated fatty acids seen in the serum of NAFLD patients[27]. Palmitate-loaded hepatocytes secreted large amounts of IL11 (28-fold higher than control, *P* < 0.0001) (Fig. 2a), produced more IL6, CCL2, and CCL5 (Fig. 2b–d), and exhibited cell death and ALT release (Fig. 2e–g).

To test if IL11 secretion from lipid-laden hepatocytes was mechanistically related to lipotoxicity we incubated cells with neutralizing anti-IL11RA antibody (X209)[7] or sgp130. X209 reduced the secretion of all cytokines, including IL11 itself, whereas sgp130 had no effect (Fig. 2a–g). This suggests the importance of autocrine loop of IL11 *cis*-signaling for hepatocyte lipotoxicity. Using hyperIL11 stimulation, which is not detected by IL11 enzyme-linked immunosorbent assay[1], we then established the existence of feed-forward autocrine IL11 signaling in hepatocytes (Supplementary Fig. 5a). The production of reactive oxygen species (ROS) from damaged mitochondria is important for lipotoxicity[23] and ROS from NOX4 are also pertinent[28]. Consistent with IL11 *cis*-driven effects on ROS in palmitate-loaded hepatocytes, we found that X209, but not sgp130, partially restored total and reduced glutathione (GSH) levels, and this was accompanied by reduction in ROS (Fig. 2h and i; Supplementary Fig. 5b).

Lipotoxicity is strongly associated with activation of JNK, which contributes to caspase-3 activation, lipoapoptosis, and dysfunctional mitochondria function. Accordingly, palmitate-loaded hepatocytes exhibited an increase in JNK activation along with ERK phosphorylation, caspase-3 cleavage, and NOX4 upregulation (Fig. 2j). Treatment of lipotoxic cells with Z-VAD-FMK prevented caspase-3 cleavage, as expected, and reduced hepatocyte death (Supplementary Fig. 5c and d). Notably, inhibition of IL11 signaling with X209 reduced palmitate-induced NOX4 upregulation along with JNK, ERK, and caspase-3 activation (Fig. 2j). While STAT3 was activated by palmitate loading, this effect appears unrelated to IL11-driven lipoapoptosis as it was unaffected by X209 (Fig. 2j).

Lipid loading of hepatocytes was confirmed by Oil Red O staining (Supplementary Fig. 5e) and quantitative analysis revealed a reduction in hepatocyte triglyceride (TG) levels with X209 treatment (Supplementary Fig. 5f). Inhibition of IL11 signaling also resulted in increased mitochondrial O2 consumption rates, and maximal and spare respiratory capacity levels (Supplementary Fig. 5g and h). This effect was mediated by IL11 *cis*-signaling as sgp130 had no effect (Supplementary Fig. 5g and h). The fact that TGs were reduced suggested increased beta-

oxidation may play a role[29]. We tested for this and found that inhibition of IL11 *cis*-signaling improved fatty acid oxidation in lipid-laden hepatocytes (Fig. 2k).

In the context of fatty liver disease, hepatocytes release factors to activate HSCs and activate/recruit immune cells, which are key events in the progression from compensated steatosis to NASH. As IL11 is secreted by lipotoxic hepatocytes (Fig. 2a), it could in theory act in paracrine on HSCs to drive HSC-to-myofibroblast transformation[7]. We cultured HSCs with conditioned media from either control or palmitate-treated hepatocytes and found that media from lipotoxic hepatocytes strongly induced ACTA2 and Collagen expression in HSCs (Fig. 2l; Supplementary Fig. 5i and j). Addition of X209 to the conditioned media blocked ACTA2 and Collagen (Fig. 2l; Supplementary Fig. 5i and j). These data demonstrate that lipotoxic hepatocytes release IL11 that acts in a paracrine fashion to activate IL11 signaling in HSCs.

**No evidence for IL11 or IL6 *trans*-signaling in two NASH models.** We then tested whether *trans*-signaling underlies NASH in vivo using two preclinical mouse NASH models: The Western Diet supplemented with fructose (WDF) model and the methionine- and choline-deficient high fat diet (HFMCD) model. The WDF model is associated with obesity, hyperlipidemia, high glucose levels, and insulin resistance and is seen as translatable to common forms of human NASH, as in diabetic patients. The HFMCD model stimulates rapid onset NASH, specifically driven by hepatocyte lipotoxicity, that is associated with weight loss in the absence of insulin resistance. Lipotoxicity is common to both models whereas obesity and insulin resistance are not.

Three weeks prior to starting either the WDF or HFMCD diet, mice were injected with an AAV8 virus encoding either albumin promoter-driven sgp130 (AAV8-Alb-sgp130), which contains the whole extracellular domain of mouse gp130 protein (amino acid 1–617), or albumin promoter alone (AAV8-Alb-Null) (Fig. 3a; Supplementary Fig. 6a, 7a). AAV8-Alb-sgp130 administration induced high levels of sgp130 in the liver, which was also detectable in the peripheral circulation, suitable for both local and systemic inhibition of putative IL6 or IL11 *trans*-signaling (Fig. 3b; Supplementary Figs. 6b, 7b and c).

After 16 weeks of WDF, IL11 levels were strongly upregulated in the liver and the periphery but IL6 expression was unaffected (Fig. 3b–d). Mice on WDF became obese (Supplementary Fig. 6c), an approximate twofold increase in liver mass and developed severe steatosis and fibrosis by gross morphology, histology, and quantitative analysis of liver triglycerides (Fig. 3e–g). These phenotypes were unaffected by high levels of sgp130 expression (Fig. 3b–g). Similarly, mice on WDF had elevated levels of ALT, AST, collagen, and peripheral cardiovascular risk factors (fasting

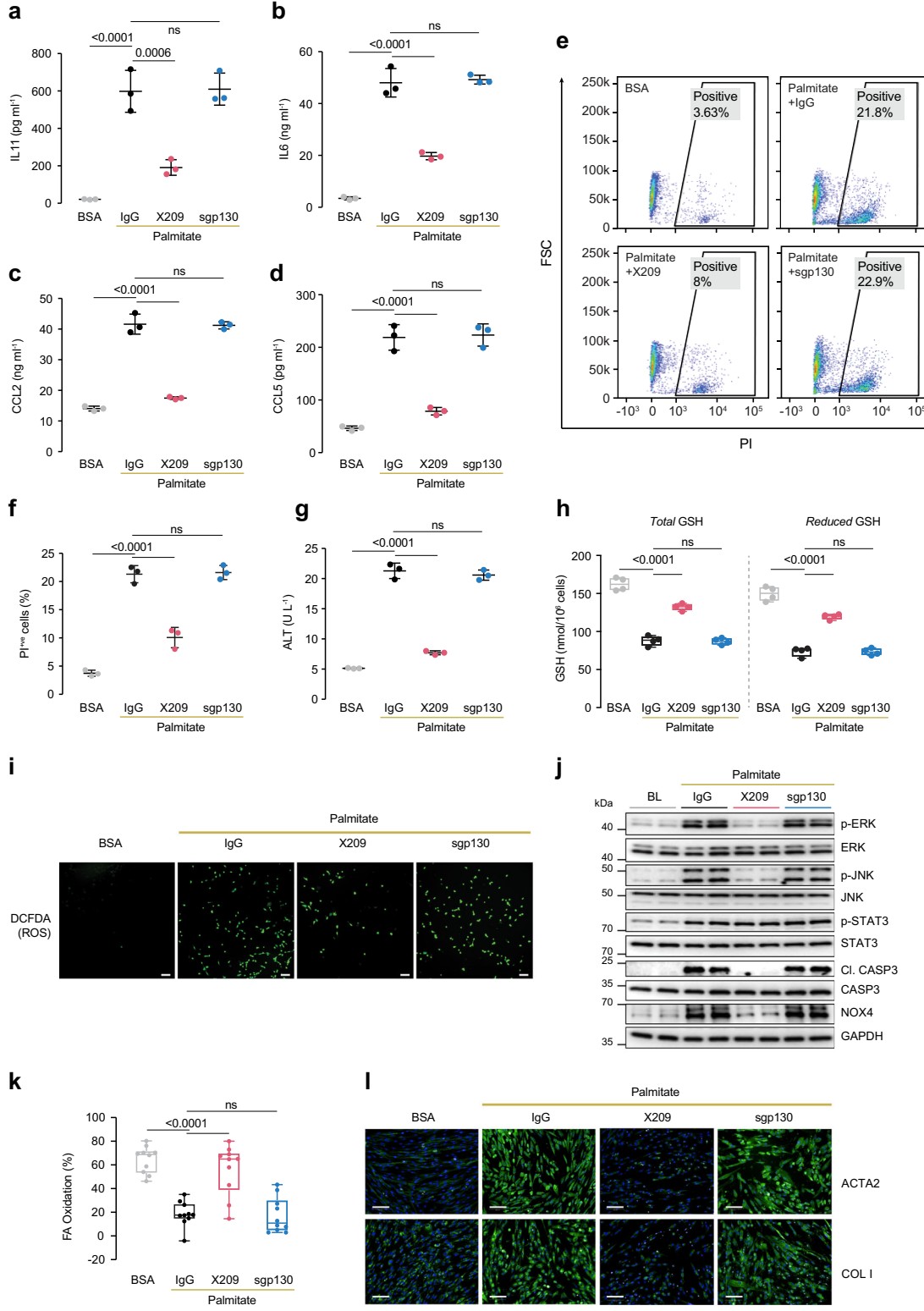

**Fig. 2 IL11 drives NASH phenotypes through autocrine effects in lipotoxic hepatocytes and paracrine activity in hepatic stellate cells. a–l** Data for palmitate (0.5 mM) loading experiment on primary human hepatocytes (24 h) in the presence of either IgG (2 μg/ml), anti-IL11RA (X209, 2 μg/ml), or sgp130 (1 μg/ml). **a** IL11, **b** IL6, **c** CCL2, and **d** CCL5 protein secretion levels as measured by ELISA of supernatants ($n = 3$). **e** Representative FSC plots and **f** quantification of PI$^{+ve}$ hepatocytes stimulated with palmitate ($n = 3$). **g** ALT levels in supernatants ($n = 3$). **h** Total and reduced hepatocyte glutathione (GSH) levels ($n = 4$). **i** Representative fluorescence images of DCFDA (2′,7′-dichlorofluorescein diacetate) staining for ROS detection (scale bars, 100 μm) ($n = 4$ independent experiments). **j** Western blots of phospho-ERK, ERK, phospho-JNK, JNK, cleaved caspase-3, caspase-3, NOX4, and GAPDH. Data from two independent biological experiments are shown. **k** Percentage of fatty acid oxidation by Seahorse assay ($n = 10$). **l** Representative fluorescence images (scale bars, 100 μm) of ACTA2$^{+ve}$ cells and Collagen I immunostaining for experiment shown in Supplementary Fig. 5j ($n = 2$ independent experiments, 14 measurements per condition per experiment). **a–d**, **f**, **g** Mean ± SD; **h**, **k** data are shown as box-and-whisker with median (middle line), 25th–75th percentiles (box), and min–max values (whiskers). **a–d**, **f–h**, **k** One-way ANOVA with Tukey's correction. Source data are provided as a Source data file.

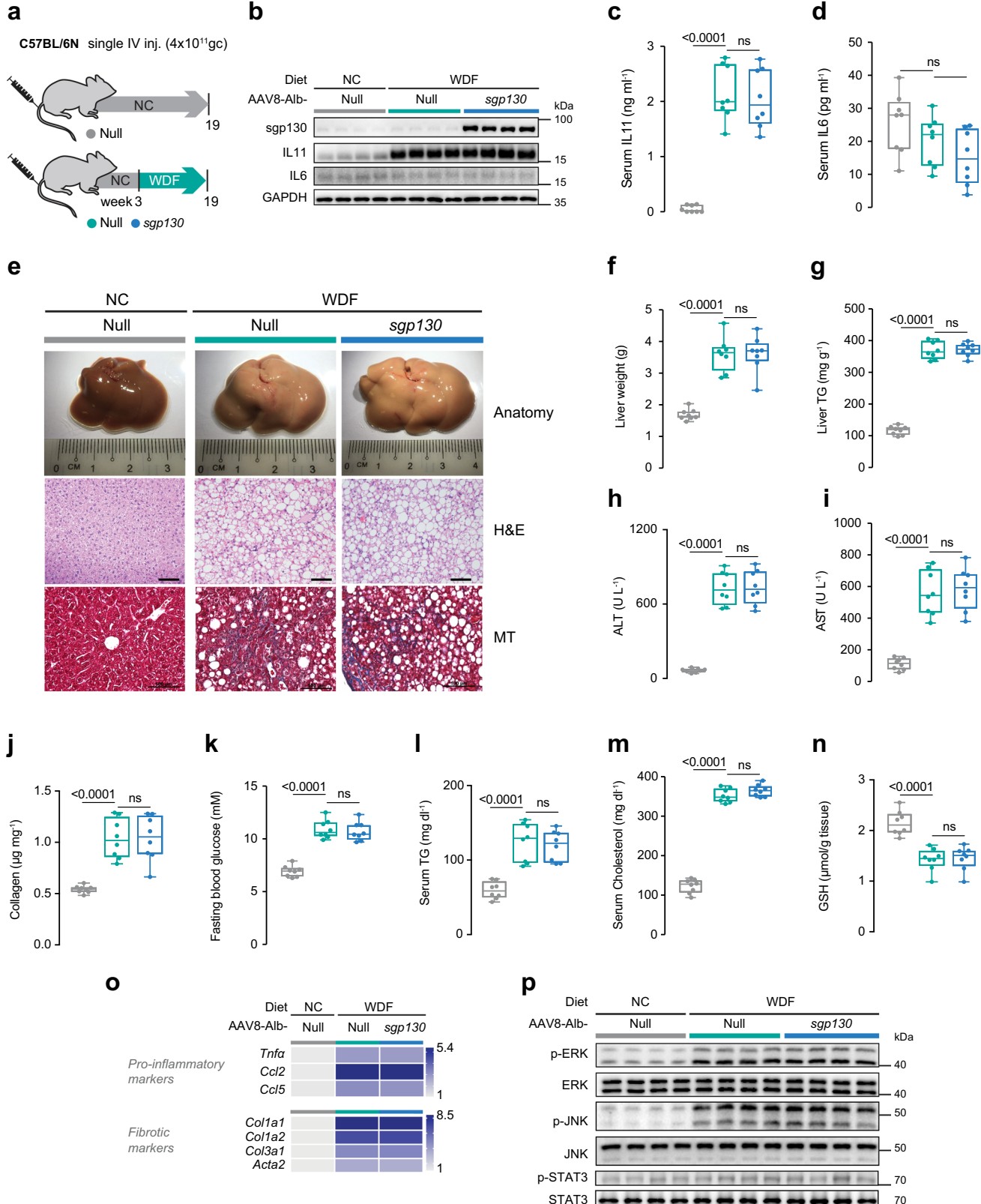

blood glucose, serum triglycerides, and serum cholesterol), along with depleted levels of GSH but none of these parameters were affected by sgp130 (Fig. 3h–n). Livers from mice on WDF showed increased expression of pro-inflammatory and fibrosis genes and this signature was unaffected by sgp130-mediated inhibition of putative *trans*-signaling (Fig. 3o; Supplementary Fig. 6d and e).

In a second set of experiments we induced NASH using the HFMCD diet (Supplementary Fig. 7a). HFMCD diet increased IL11 levels in liver and serum, whereas IL6 levels were slightly lower in the liver albeit mildly increased in the periphery (Supplementary Fig. 7b, d and e). Mice on HFMCD diet developed rapid and profound steatosis by gross morphology,

**Fig. 3 Inhibition of IL6 family cytokine *trans*-signaling has no effect on NASH or metabolic phenotypes in mice on Western Diet supplemented with fructose. a** Schematic of WDF feeding in mice with hepatocyte-specific expression of sgp130 for data shown in (**b**–**p**). Three weeks following AAV8-Alb-Null or AAV8-Alb-sgp130 virus injection, mice were fed WDF for 16 weeks. **b** Western blots showing hepatic levels of sgp130, IL11, IL6, and GAPDH as internal control (*n* = 4 mice/group). **c** Serum IL11 levels. **d** Serum IL6 levels. **e** Representative gross anatomy, H&E-stained (scale bars, 50 μm), and Masson's Trichrome (scale bars, 100 μm) images of livers. Representative dataset from *n* = 8 mice/group is shown for gross anatomy; representative dataset from *n* = 4 mice/group is shown for H&E-stained and Masson's Trichrome images. **f** Liver weight. **g** Hepatic triglycerides content. **h** Serum ALT levels. **i** Serum AST levels. **j** Hepatic collagen levels. **k** Fasting blood glucose levels. **l** Serum triglycerides levels. **m** Serum cholesterol levels. **n** Hepatic GSH content. **o** Hepatic pro-inflammatory and fibrotic genes expression heatmap (values are shown in Supplementary Fig. 6d and e). **p** Western blots of hepatic phospho-ERK, ERK, phospho-JNK, JNK, phospho-STAT3, and STAT3 (*n* = 4 mice/group). **c**, **d**, **f**–**o** *n* = 8 mice/group. **c**, **d**, **f**–**n** Data are shown as box-and-whisker with median (middle line), 25th–75th percentiles (box), and min–max values (whiskers); one-way ANOVA with Tukey's correction. Source data are provided as a Source data file.

histology, and molecular assays, which was unaltered by sgp130 expression (Supplementary Fig. 7f and g). Hepatocyte damage markers (ALT and AST) and collagen expression were elevated and GSH levels were depleted by HFMCD diet, irrespective of sgp130 expression (Supplementary Fig. 7f, h–k). Similarly, the HFMCD diet was associated with dysregulated expression of inflammation and fibrosis genes and these molecular phenotypes were unaffected by sgp130 expression (Supplementary Fig. 7l and m).

At the signaling level, both WDF and HFMCD diets stimulated ERK and JNK activation, consistent with elevated IL11 *cis*-signaling (Fig. 3p; Supplementary Fig. 7n). In contrast, phospho-STAT3 levels in the liver were not elevated by WDF (Fig. 3p) and were mildly increased in mice on the HFMCD diet (Supplementary Fig. 7n). In all instances, there was no effect of sgp130 on diet-induced signaling events. Overall, these data suggest that neither IL6 nor IL11 *trans*-signaling plays a role in NASH, which is consistent with other studies where IL6 family *trans*-signaling has not been detected[20,21,30,31].

**Hepatocyte-specific IL11 *cis*-signaling is required to initiate NASH.** While we found no evidence to support IL11 *trans*-signaling in NASH models, our in vitro data showed evidence of pathological IL11 *cis*-signaling in lipotoxic hepatocytes. To test the effects of IL11 *cis*-signaling in heptocytes in vivo, we administered AAV8-Alb-Cre to *Il11ra1*<sup>loxP/loxP</sup> mice to delete *Il11ra1* specifically in hepatocytes (CKO mice). CKO mice were then fed either normal chow (NC), HFMCD diet or WDF (Figs. 4a, 5a). Liver IL11RA protein was greatly diminished in the CKOs following AAV8-Alb-Cre injection, showing the model to be effective and suggesting that hepatocytes are the largest hepatic reservoir of IL11RA (Figs. 4b, 5b). Both WT and CKO mice had similar levels of serum IL11 after 4 weeks of HFMCD and 16 weeks of WDF (Supplementary Figs. 8a, 9a).

In addition to rapidly stimulating lipotoxicity-driven NASH, the HFMCD diet causes weight loss[32]. Surprisingly, weight loss in mice on the HFMCD diet was initially limited and later reversed in CKO mice (Fig. 4c; Supplementary Fig. 8b). Mice on WDF gained weight and fat mass throughout the experimental period, as expected. However, and equally surprising, these obesity phenotypes were mitigated in CKO mice (Fig. 5c and d; Supplementary Fig. 9b). These data suggest that inhibition of IL11 signaling is permissive for weight homeostasis, with context-specific anti-cachectic or anti-obesity effects, which requires further study.

By gross morphology, histology and quantitative triglyceride analysis, the CKO mice on either HFMCD or WDF diet were protected from steatosis (Figs. 4d and e, 5e, and f) and those on WDF had less liver mass (Fig. 5g). Liver damage markers were markedly reduced in CKO mice fed with either HFMCD diet (reduction: ALT, 99%; AST, 97%; *P* < 0.0001 for both) or WDF (reduction: ALT, 98%; AST, 98%; *P* < 0.0001 for both) and found

to be comparable to NC control levels (Figs. 4f and g, 5h and i). In both models, GSH levels were diminished in WT mice on both NASH diets but normalized in CKOs (Figs. 4h, 5j).

Liver fibrosis was greatly reduced in CKO mice on either NASH diet as compared to WT (reduction: HFMCD, 87%; WDF, 64%) (Figs. 4d, 4i, 5e, 5k). Upregulation of pro-inflammatory and fibrosis genes in mice on either the HFMCD or WDF diets was also diminished in the CKOs (Figs. 4j, 5l; Supplementary Figs. 8c and d, 9c and d). This suggests that transformation of HSCs to myofibroblasts and activation of immune cells are, in part, secondary to upstream, IL11-driven events in hepatocytes that are consistent with the paracrine effects we detected in vitro (Fig. 2l; Supplementary Fig. 5i and j). At the signaling level, both HFMCD diet and WDF resulted in elevated ERK and JNK phosphorylation. This was prevented in CKO mice, consistent with inhibition of IL11 signaling in hepatocytes (Figs. 4k, 5m).

Mice on WDF are known to develop hyperglycemia, hypertriglyceridemia, and hypercholesterolemia, all of which were improved in the CKOs (Supplementary Fig. 9e–g). Furthermore, as compared to controls, CKOs on WDF had elevated serum levels of β-hydroxybutyrate, a peripheral marker of liver fatty acid oxidation and ketone production (Supplementary Fig. 9h). Overall these data suggest improvement of liver metabolism in CKOs, in keeping with our finding that inhibition of IL11 signaling promotes fatty acid oxidation in lipotoxic hepatocytes (Fig. 2k).

**Reconstitution of IL11 *cis*-signaling in hepatocytes in *IL11ra1* null mice restores steatohepatitis but not liver fibrosis.** To complement our loss-of-function experiments using the CKO mice we employed in vivo gain-of-function experiments. To do so, we assessed whether restoring IL11 *cis*- or *trans*-signaling specifically in hepatocytes in mice with global *Il11ra1* deletion (*Il11ra1*<sup>−/−</sup> knockouts (KOs)) resulted in disease. KO mice were injected with AAV8 encoding either the full-length, membrane-bound *Il11ra1* (*mbIl11ra1*; to reconstitute *cis*-signaling) or a secreted/soluble form of *Il11ra1* (*sIl11ra1*, which constitutes the extracellular portion of *Il11ra1*; to enable *trans*-signaling) or a control construct, and the animals were then fed with NC, HFMCD diet, or WDF (Fig. 6a; Supplementary Fig. 10a, 11a).

KO mice injected with AAV8-Alb-*mbIl11ra1* re-expressed IL11RA1 on hepatocytes and KO mice injected with AAV8-Alb-*sIl11ra1* had increased expression of sIL11RA1 in both the liver and the periphery (Fig. 6b; Supplementary Fig. 10b, 11b and c). As expected, WT mice receiving control AAV8 constructs (AAV8-Alb-Null) on NC had normal livers and developed steatosis, inflammation, liver damage, and liver fibrosis when on either HFMCD diet or WDF (Fig. 6c–j; Supplementary Fig. 10c and d, 11d–k). KO mice injected with control virus and fed either HFMCD or WDF diets were protected from NASH phenotypes, although protection from NASH with germline global deletion of

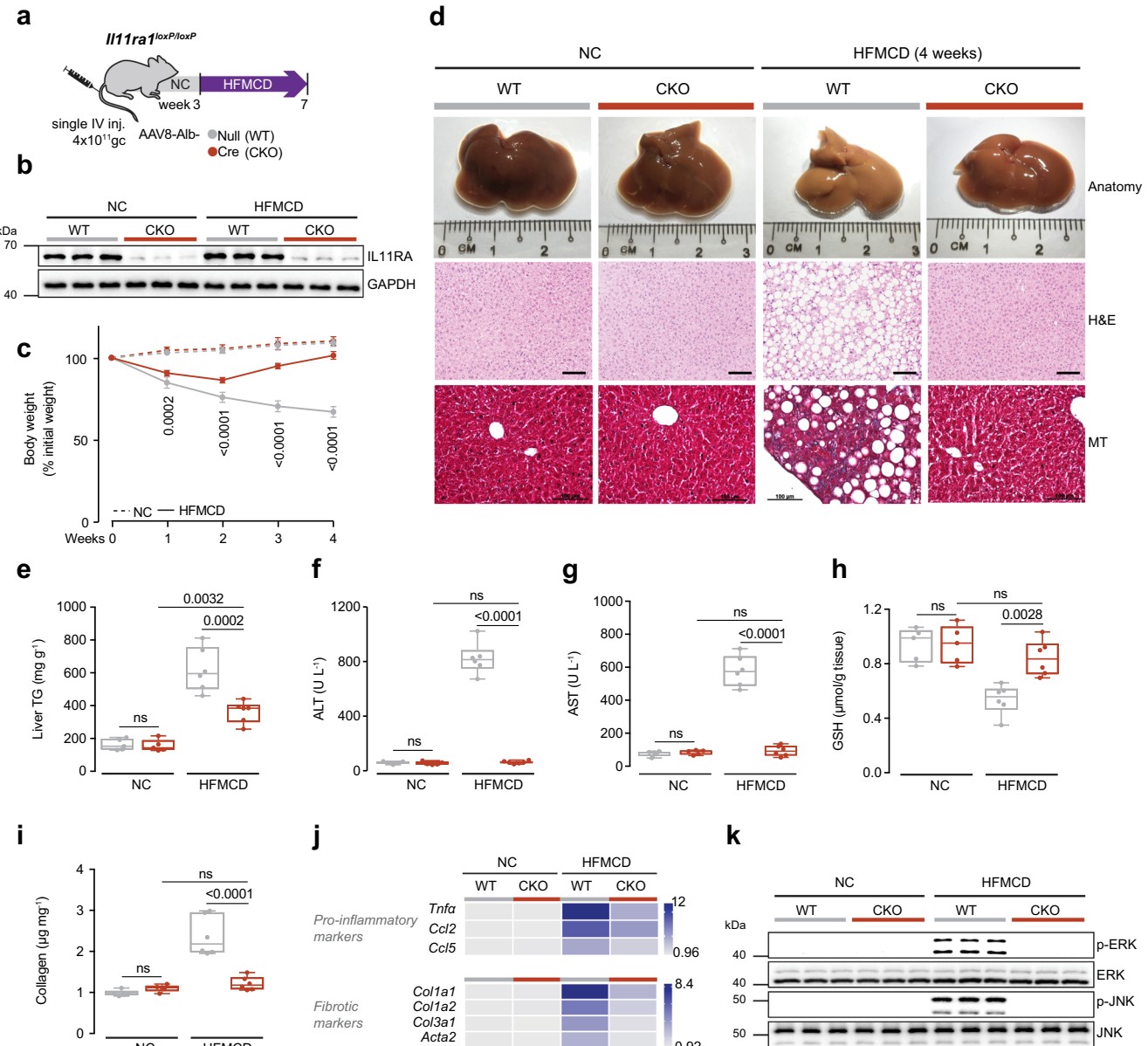

**Fig. 4 Hepatocyte-specific inhibition of IL11 *cis*-signaling protects mice against HFMCD diet-induced NASH. a** Schematic of HFMCD feeding regimen for AAV8-Alb-Cre injected *Il11ra1*[loxP/loxP] (conditional knockout; CKO) mice for experiments shown in (**b–k**). *Il11ra1*[loxP/loxP] mice were intravenously injected with either AAV8-Alb-Null or AAV8-Alb-Cre to delete *Il11ra1* specifically in hepatocytes 3 weeks prior to the start of HFMCD diet. **b** Western blots of hepatic IL11RA and GAPDH (*n* = 3 mice/group). **c** Body weight (shown as a percentage (%) of initial body weight). **d** Representative gross anatomy, H&E-stained (scale bars, 50 μm), and Masson's Trichrome (scale bars, 100 μm) images of livers. Representative dataset from *n* = 5 mice/group is shown for gross anatomy; representative dataset from *n* = 4 mice/group is shown for H&E-stained and Masson's Trichrome images. **e** Hepatic triglycerides content. **f** Serum ALT levels. **g** Serum AST levels. **h** Hepatic GSH content. **i** Hepatic collagen levels. **j** Heatmap showing hepatic mRNA expression of pro-inflammatory markers (*Tnfα, Ccl2, Ccl5*) and fibrotic markers (*Col1a1, Col1a2, Col3a1, Acta2*). Values are shown in Supplementary Fig. 8c and d. **k** Western blots showing hepatic ERK and JNK activation status (*n* = 3 mice/group). **c, e–j** NCD (*n* = 5 mice/group), HFMCD (*n* = 6 mice/group). **c** Data are shown as mean ± SD, two-way ANOVA with Tukey's correction, statistical significance (*P* values) are shown for comparison between WT HFMCD and CKO HFMCD; **e–i** data are shown as box-and-whisker with median (middle line), 25th–75th percentiles (box), and min–max values (whiskers); two-way ANOVA with Tukey's correction. Source data are provided as a Source data file.

*Il11ra1* was not as strong as seen in the CKOs (Figs. 4d–j, 5c–l, 6c–j; Supplementary Fig. 8c and d, 9b–g, 10c and d, 11d–k).

Restoration of IL11 *cis*-signaling in KO mice using mbIl11ra1 recapitulated hepatic steatosis and inflammation that was evident from gross morphology to molecular patterns of gene expression and signaling (Fig. 6c–h, j and k; Supplementary Fig. 10c, 11d–h, 11j, 11l). However, hepatic collagen content and fibrotic gene expression were not restored (Fig. 6c, i and j; Supplementary

Fig. 10d, 11d, 11i, 11k) presumably because IL11 *cis*-signaling in HSCs, important for HSC-to-myofibroblast transformation[7], was unaffected by the albumin-driven Il11ra1 expression (i.e., HSCs remain deleted for *Il11ra1* in this model).

In contrast, expression of the sIL11RA in hepatocytes of KOs, which would theoretically activate *trans*-signaling as IL11 levels are already elevated, had no effect despite very high IL11RA levels (Fig. 6b; Supplementary Fig. 10b, 11b and c) and mice remained

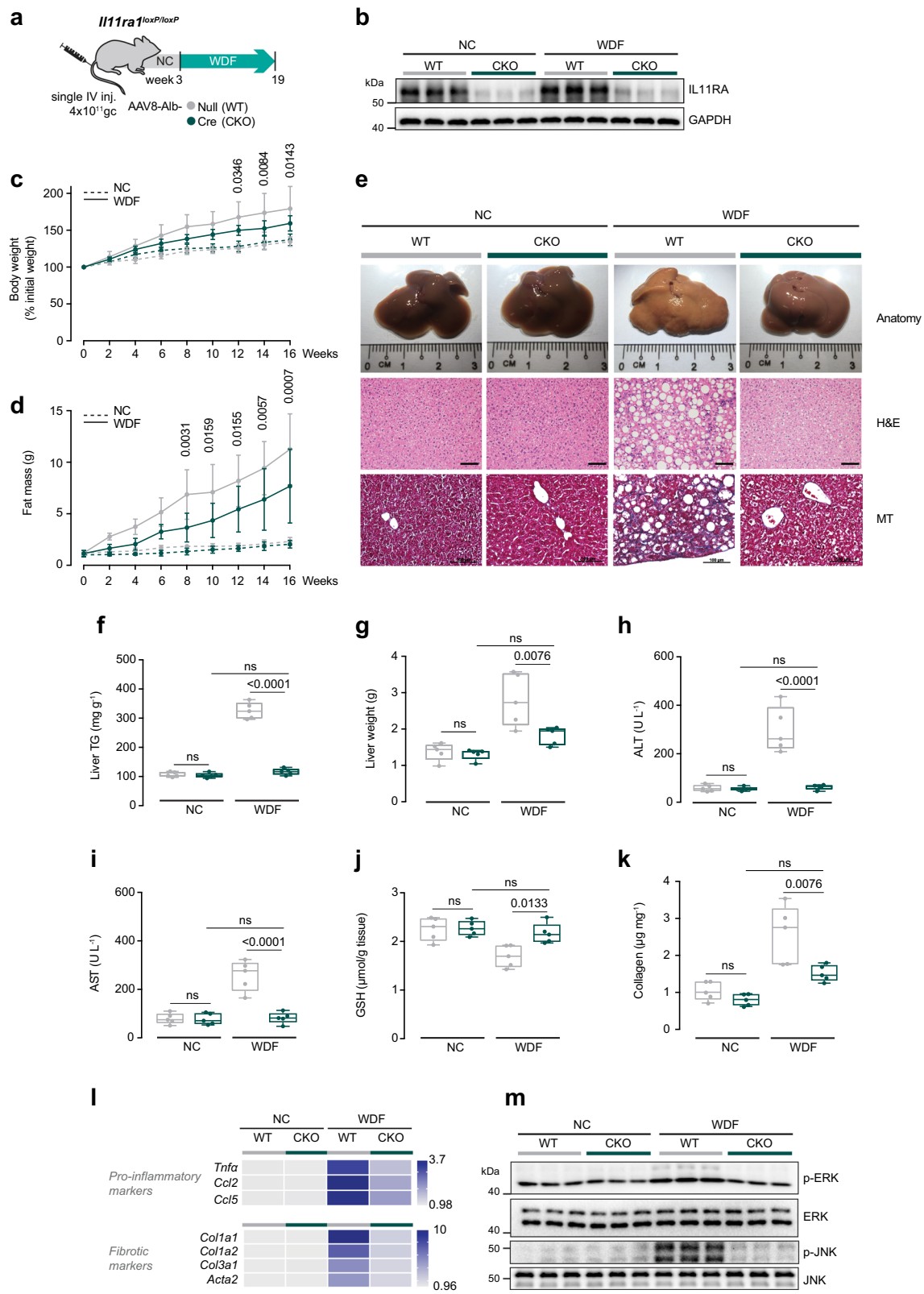

protected from NASH (Fig. 6c–j; Supplementary Fig. 10c, 10d, 11d–k). Signaling changes were consistent in that mIL11RA expression restored pathological ERK and JNK activation in KOs on either diet, whereas sIL11RA1 did not (Fig. 6k; Supplementary Fig. 11l). Similarly, in the WDF model, restoration of hepatocyte-specific IL11 *cis*-signaling (mIL11RA) in KO mice caused

hyperglycemia, hypertriglyceridemia, and hypercholesterolemia but expression of sIl11ra1 did not (Fig. 6l–n).

## Discussion

Metabolic liver disease commonly occurs in the context of obesity and type 2 diabetes and manifests initially as NAFLD that can

**Fig. 5 Hepatocyte-specific inhibition of IL11 *cis*-signaling protects mice against WDF-induced obesity and NASH. a** Schematic of WDF-fed control and CKO mice for data shown in (**b–m**). Three weeks following AAV8-Alb-Null or AAV8-Alb-Cre virus injection, CKO mice were fed WDF for 16 weeks. **b** Western blots showing hepatic levels of IL11RA and GAPDH (*n* = 3 mice/group). **c** Body weight (shown as a percentage (%) of initial body weight). **d** Fat mass. **e** Representative gross anatomy, H&E-stained (scale bars, 50 μm), and Masson's Trichrome (scale bars, 100 μm) images of livers. Representative dataset from *n* = 5/group is shown for gross anatomy; representative dataset from *n* = 4 mice/group is shown for H&E-stained and Masson's Trichrome images. **f** Hepatic triglycerides content. **g** Liver weight. **h** Serum ALT levels. **i** Serum AST levels. **j** Hepatic GSH content. **k** Hepatic collagen levels. **l** Hepatic pro-inflammatory and fibrotic genes expression on heatmap (values are shown in Supplementary Fig. 9c and d). **m** Western blots showing activation status of hepatic ERK and JNK (*n* = 3 mice/group). **c, d, f–l** *n* = 5 mice/group. **c, d** Data are shown as mean ± SD, two-way ANOVA with Tukey's correction, statistical significance (*P* values) are shown for comparison between WT WDF and CKO WDF; **f–k** data are shown as box-and-whisker with median (middle line), 25th–75th percentiles (box), and min–max values (whiskers); two-way ANOVA with Tukey's correction. Source data are provided as a Source data file.

progress to NASH[22,33]. A key underlying pathology in the progression to NASH is "substrate overload", whereby a nutritional abundance overruns the hepatocyte's ability to process fat, causing lipotoxicity. Cytokines are key NASH factors secreted from lipotoxic hepatocytes[22] and here we establish IL11 as an important component of the lipotoxic milieu and a driver of the NAFLD-to-NASH transition.

A large body of evidence supports the idea that IL6 *cis*- and *trans*-signaling in the liver is beneficial[16,17,19,25]. However, at the same time, a pathogenic role for IL6 *trans*-signaling in hepatic steatosis has also been proposed[30,34]. Using man-made, artificial protein constructs we found that IL11 *trans*-signaling is toxic in hepatocytes, whereas hyperIL6 appears protective. However, we found no evidence to support a meaningful role for IL6 or IL11 *trans*-signaling in a biologically relevant context either in vitro or in vivo, using both gain- and loss-of-function approaches. Notably, sgp130, a therapeutic agent that inhibits IL6 trans-signaling, had no effect on lipotoxicity, NAFLD or NASH. Thus, we suggest that IL6 family member *trans*-signaling has no role in hepatocytes or NASH, which is in agreement with studies outside the liver[20,21].

Our data show a central importance of IL11 *cis*-signaling in hepatocytes for multiple NASH pathologies. This effect was established using both hepatocyte-specific loss-of-function on a wild-type genetic background and also hepatocyte-specific gain-of-function on an *Il11ra1* null background. This overturns the suggestion in the literature that IL11 is protective for hepatocytes based on the use of recombinant human IL11 in murine models of liver disease[8,10–12]. While restoration of IL11 *cis*-signaling in hepatocytes causes steatohepatitis in mice globally deleted for *Il11ra1*, lipotoxicity-associated fibrosis was still prevented. In contrast, hepatocyte-specific *Il11ra1* deletion protected mice from both steatohepatitis and fibrosis. This places hepatocyte dysfunction upstream of HSC activation, which is consistent with the observed paracrine effects of IL11 from lipotoxic hepatocytes.

We repeatedly documented metabolically advantageous effects associated with the inhibition of IL11 signaling. In vitro, inhibition of IL11 improved mitochondrial function and increased beta-oxidation along with a reduction of intracellular triglycerides and ROS production. The effects on ROS are likely complex as IL11 induces NOX4 but also impacts mitochondria function and perhaps also the endoplasmic reticulum, directly or indirectly. Interestingly, inhibition of NOX4 was more effective than caspase inhibition in reducing IL11-stimulated cell death. Thus IL11-induced NOX4/ROS lies upstream of late-stage ERK and caspase activation in hepatocytes and apoptosis is not the only form of cell death in this context. In vivo, deletion of *Il11ra1* in hepatocytes limited WDF-induced fat accumulation and body weight gain while reducing serum glucose, triglyceride, and cholesterol levels. This was associated with lower liver fat, lesser hepatic oxidative stress, and increased serum levels of beta-hydroxybutyrate, thought metabolically beneficial in itself[35].

Our studies have limitations and pose questions. The published literature suggests IL6R is highly expressed in hepatocytes[16] and it was surprising that primary human hepatocytes express very little IL6R. The differential activation of ERK/JNK by IL11 as compared to STAT3 by IL6 at 24 h in hepatocytes was also notable but the underlying mechanisms are unknown. While we show consistent effects of IL11 inhibition on pro-inflammatory factors we did not specifically address effects on immune cells themselves. Metabolic effects appear closely related to IL11-mediated NOX4-derived ROS but ROS from other sources also likely contributes. Temporal relationships between ERK and NOX4 activation require further evaluation. We found a beneficial effect of hepatocyte-specific IL11 inhibition on reducing fat deposition and body weight gain in mice on an obesogenic NASH diet, which while notable was not studied in depth. We surmise that IL11 hepatocyte biology is a nascent field and that these various matters require further study.

In conclusion, we propose a model for lipotoxicity-driven NAFLD-to-NASH transitions whereby lipid-laden hepatocytes secrete IL11 leading to autocrine hepatocyte metabolic dysfunction and cell death along with paracrine activation of neighbouring HSCs and other cells (Fig. 7). We rule out IL6 or IL11 *trans*-signaling as relevant for hepatocyte biology or liver pathology in lipotoxicity. We suggest that inhibiting IL11 signaling in hepatocytes targets an initiating nexus for diet-induced steatohepatitis that impacts subsequent liver fibrosis and inflammation. Hence, therapeutic inhibition of IL11-induced lipotoxicity may be beneficial in metabolic liver diseases, such as NASH.

## Methods

**Ethics statements**. All experimental protocols involving human subjects (commercial primary human cell lines and human liver sections) have been performed in accordance with the *ICH Guidelines for Good Clinical Practice*. As written in their respective datasheets, ethical approvals have been obtained by the relevant parties and all participants gave written informed consent: commercial human liver sections (by Abcam); liver sections from healthy control and NASH patients (by Fibrofind); primary human hepatocytes and hepatic stellate cells (by ScienCell); HepG2 and THP-1 (by ATCC).

Animal studies were carried out in compliance with the recommendations in the *Guidelines on the Care and Use of Animals for Scientific Purposes* of the *National Advisory Committee for Laboratory Animal Research* (NACLAR). All experimental procedures were approved (SHS/2014/0925 and SHS/2019/1482) and conducted in accordance with the SingHealth Institutional Animal Care and Use Committee.

**AAV8 vectors**. All Adeno-associated virus serotype 8 (AAV8) vectors used in this study were synthesized by Vector Biolabs. AAV8 vector carrying a mouse membrane-bound *Il11ra1* cDNA (NCBI accession number: BC069984), a mouse soluble *Il11ra1* cDNA, and a mouse soluble *gp130* cDNA driven by Albumin (*Alb*) promoter is referred to as AAV8-Alb-mbIl11ra1, AAV8-Alb-sIl11ra1, and AAV8-Alb-sgp130, respectively. AAV8-Alb-sgp130 and AAV8-Alb-sIl11ra1 were constructed by removing the transmembrane and cytoplasmic regions of mouse *gp130* sequence (NCBI accession number: BC058679) and mouse *Il11ra1* sequence, respectively. AAV8-Null vector was used as vector control. To specifically delete

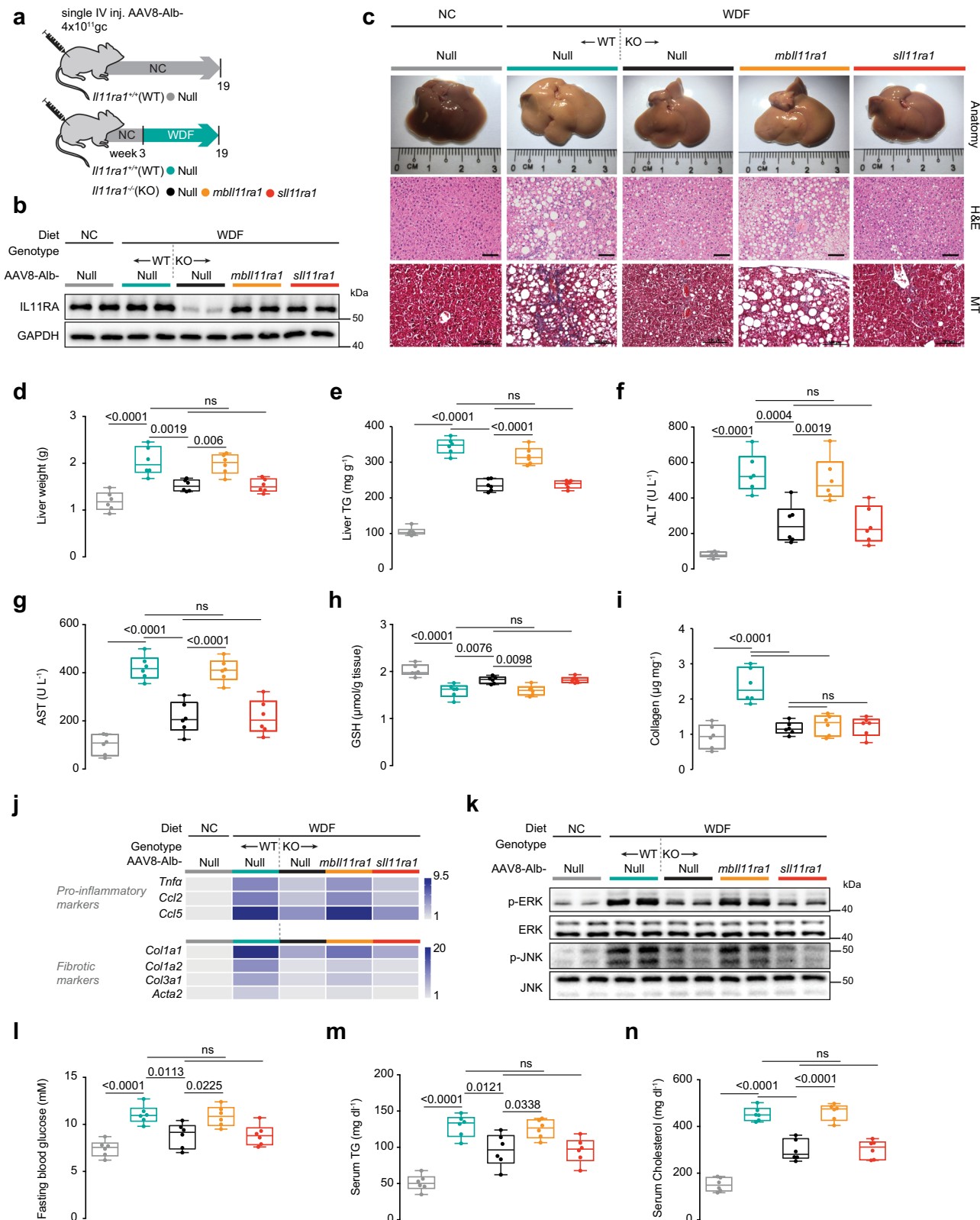

*Il11ra1* in Albumin-expressing cells, AAV8-Alb-iCre vector was injected to mice homozygous for LoxP-flanked *Il11ra1* alleles (*Il11ra1^loxP/loxP* mice).

**Antibodies**. ACTA2 (ab7817, Abcam, 1:1000 for western blot and 1:500 for operetta assay), Albumin (ab207327, Abcam, 1:100 for IF and flow cytometry), Cleaved Caspase-3 (9664, CST, 1:1000), Caspase-3 (9662, CST, 1:1000), Collagen I (ab34710, Abcam, 1:500), phospho-ERK1/2 (4370, CST, 1:1000), ERK1/2 (4695, CST, 1:1000), GAPDH (2118, CST, 1:1000), gp130 (human, PA5-28932, Thermo

Fisher, 1:100), gp130 (mouse, PA5-99526, Thermo Fisher, 1:100), gp130 (extra-cellular, PA5-77476, Thermo Fisher, 1:1000), IgG (11E10, Aldevron), IL6 (AF506, R&D Systems, 1:1000), IL6R (flow cytometry, ab222101, Abcam, 1:100), IL6R (human, IHC and IF, MA1-80456, Thermo Fisher, 1:100), IL6R (mouse, ab83053, Abcam, 1:100), IL11 (X203, Aldevron), IL11RA (inhibition study, X209, Aldevron), IL11RA (IHC, IF, flow cytometry, ab125015, Abcam, 1:100), IL11RA (western blot, sc-130920, Santa Cruz, 1:1000), phospho-JNK (4668, CST, 1:1000), JNK (9252, CST, 1:1000), NOX4 (MA5-32090, Invitrogen, 1:1000), phospho-STAT3 (4113, CST, 1:1000), STAT3 (4904, CST, 1:1000), mouse Alexa Fluor 488 secondary

**Fig. 6 Hepatocyte-specific IL11 *cis*-signaling but not IL11 *trans*-signaling drives steatohepatitis in mice on WDF. a** Schematic showing WDF feeding regimen of *Il11ra1*$^{+/+}$ (WT) and *Il11ra1*$^{-/-}$ (KO) mice for experiments shown in (**b-n**). AAV8-Alb-Null, AAV8-Alb-mbIl11ra1 (full-length membrane-bound Il11ra1), and AAV8-Alb-sIl11ra1 (soluble form of Il11ra1)-injected KO mice were given 16 weeks of WDF feeding, three weeks following virus administration. **b** Western blots showing hepatic levels of IL11RA and GAPDH ($n = 2$ mice/group). **c** Representative gross anatomy, H&E-stained (scale bars, 50 μm) and Masson's Trichrome (scale bars, 100 μm) images of livers. Representative dataset from $n = 6$ mice/group is shown for gross anatomy; representative dataset from $n = 4$ mice/group is shown for H&E-stained and Masson's Trichrome images. **d** Liver weight. **e** Hepatic triglycerides content. **f** Serum ALT levels. **g** Serum AST levels. **h** Hepatic GSH content. **i** Hepatic collagen content. **j** Hepatic pro-inflammatory and fibrotic genes expression heatmap (values are shown in Supplementary Fig. 10c and d). **k** Western blots showing activation status of hepatic ERK and JNK ($n = 2$ mice/group). **l** Fasting blood glucose levels. **m** Serum triglycerides levels. **n** Serum cholesterol levels. **d-j, l-n** $n = 6$ mice/group. **d-i, l-n** Data are shown as box-and-whisker with median (middle line), 25th–75th percentiles (box), and min–max values (whiskers); one-way ANOVA with Tukey's correction. Source data are provided as a Source data file.

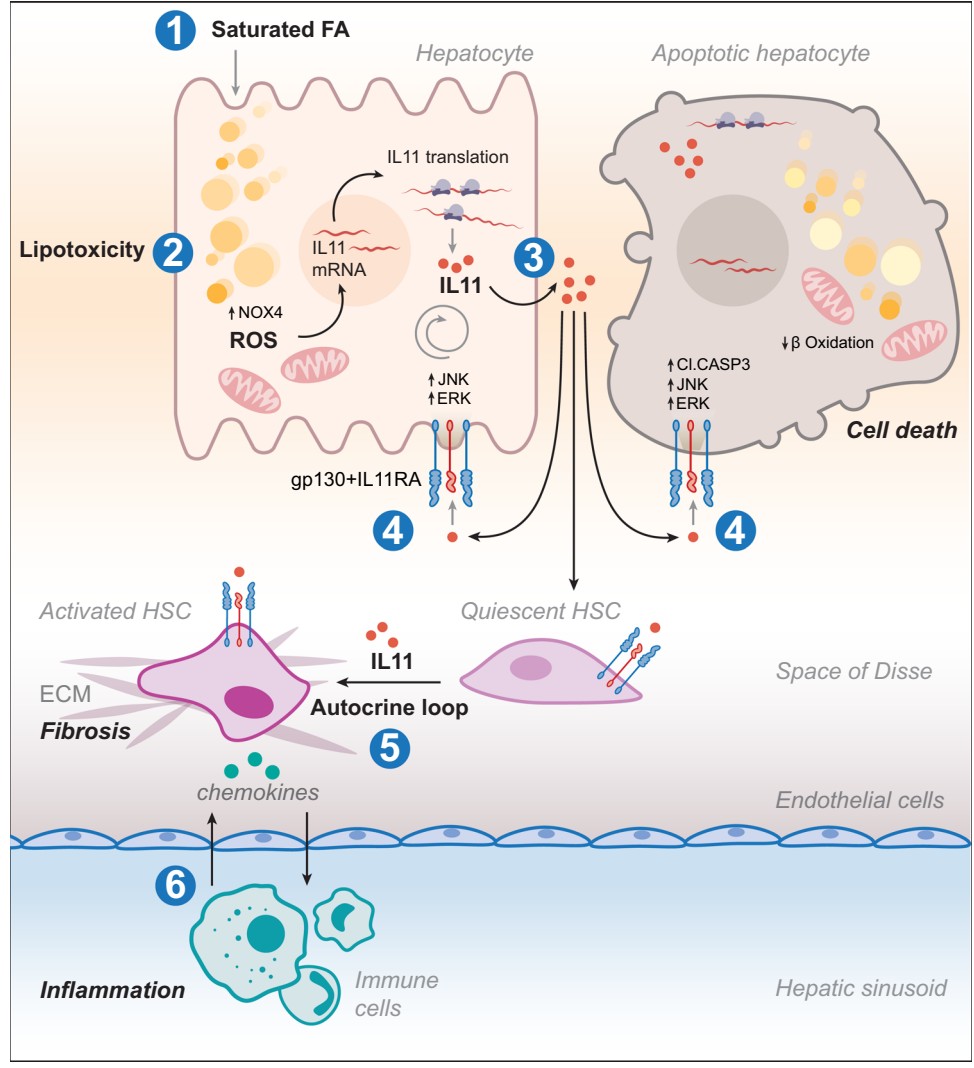

**Fig. 7 Proposed mechanism for IL11 in lipotoxicity-driven NASH transition.** Excessive lipid accumulation in hepatocytes stimulates IL11 protein secretion and autocrine IL11 activity, which upregulates NOX4 and increases reactive oxygen species production. Subsequently, hepatocyte mitochondrial oxidative capacity and fatty acid metabolism impaired and steatosis established. ERK, JNK, and caspase-3 become activated and this leads to lipoapoptosis, along with other forms of cell death. IL11 also acts in paracrine to drive hepatic stellate cell-to-myofibroblast transformation and fibrosis. Cytokines and chemokines released from lipotoxic hepatocytes and HSCs activate and recruit immune cells causing inflammation.

antibody (ab150113, Abcam, 1:200), mouse HRP (7076, CST, 1:2000), rabbit Alexa Fluor 488 secondary antibody (ab150077, Abcam, 1:200), rabbit HRP (7074, CST, 1:2000), rat Alexa Fluor 488 secondary antibody (ab150157, Abcam, 1:200), rat HRP (31470, Santa Cruz, 1:800).

**Recombinant proteins.** Commercial recombinant proteins: Human hyperIL6 (IL6R:IL6 fusion protein, 8954-SR, R&D Systems), human IL6 (206-IL-010, R&D Systems), human soluble gp130 Fc (671-GP-100, R&D Systems), human IL11RA (8895-MR-050, R&D Systems).

Custom recombinant proteins: Human IL11 (UniProtKB:P20809, Genscript). Human hyperIL11 (IL11RA:IL11 fusion protein), which mimics the *trans*-signaling complex, was constructed using a fragment of IL11RA (amino acid residues 1–317; UniProtKB: Q14626) and IL11 (amino acid residues 22–199, UniProtKB: P20809) with a 20 amino acid linker (GPAGQSGGGGGSGGGSGGGSV)[1].

**Chemicals.** 4′,6-diamidino-2-phenylindole (DAPI, D1306, Thermo Fisher), diphenyleneiodonium chloride (DPI, 141310, Abcam), GKT-137831 (17764, Cayman Chemical), palmitate (P5585, Sigma), paraformaldehyde (PFA, 28908;

Thermo Fisher), phorbol 12-myristate 13-acetate (PMA, P1585, Sigma), Triton X-100 (T8787, Sigma), and Z-VAD-FMK (FMK001, Sigma).

**Cell culture**. All the experiments performed with primary human hepatocytes, primary adult mouse hepatocytes, and primary adult human hepatic stellate cells were carried out at low cell passage (≤P3).

1.  Primary human hepatocytes culture
    Primary human hepatocytes (5200, ScienCell) were maintained in hepatocyte medium (5201, ScienCell) supplemented with 2% fetal bovine serum, 1% penicillin-streptomycin at 37 °C and 5% $CO_2$. Hepatocytes were serum-starved overnight unless otherwise specified in the methods prior to 24 h stimulation with different doses of various recombinant proteins as outlined in the main text and/or figure legends.
2.  Primary adult mouse hepatocytes culture
    Mouse hepatocytes (ABC-TC3928, AcceGen Biotech) were maintained in mouse hepatocyte medium (ABC-TM3928, AcceGen Biotech) supplemented with 1% penicillin-streptomycin.
3.  Primary adult human hepatic stellate cells
    HSCs (5300, ScienCell) were cultured in stellate cells complete media (5301, ScienCell) on poly-L-lysine-coated plates (2 µg/cm², 0403, ScienCell). HSCs were serum-starved overnight prior to 24 h stimulation with conditioned media from BSA or palmitate-stimulated hepatocyte (24 h) in the presence of various recombinant proteins as outlined in the main text and/or figure legends.
4.  HepG2 culture
    HepG2 (ATCC) were cultured in Eagle's minimum essential medium (30-2003, ATCC) supplemented with 10% FBS.
5.  AML12 culture
    AML12 (ATCC) were cultured in DMEM:F12 medium (30-2006, ATCC) supplemented with 10% FBS, 10 µg/ml insulin, 5.5 µg/ml transferrin, 5 ng/ml selenium, and 40 ng/ml dexamethasone.
6.  THP-1 culture
    THP-1 (ATCC) were cultured in RPMI 1640 (A1049101, Thermo Fisher) supplemented with 10% FBS and 0.05 mM β-mercaptoethanol. THP-1 cells were differentiated with 10 ng/ml of PMA in RPMI 1640 for 48 h.

**Palmitate (saturated fatty acid) treatment in vitro**. Palmitate (0.5 mM) conjugated in fatty acids free BSA in the ratio of 6:1 was used to treat cells as described in figure legends; 0.5% BSA solution was used as control.

**Flow cytometry**. For surface IL11RA, IL6R, and gp130 analysis, primary human hepatocytes and THP-1 cells were stained with IL11RA, IL6R, or gp130 antibody, and the corresponding Alexa Fluor 488 secondary antibody. Omission of primary antibody staining was used as negative control. Cell death analysis was performed by staining primary human hepatocytes with Dead Cell Apoptosis Kit with Annexin V FITC and PI (V13242, Thermo Fisher). PI[+ve] cells were then quantified with the flow cytometer (Fortessa, BD Biosciences) and analyzed with FlowJo version X software (TreeStar): the preliminary FSC/SSC gates of the starting cell population was 10,000 events. Debris (SSC-A vs FSC-A) and doublets (FSC-H vs FSC-A) were excluded. Boundaries between "positive" and "negative" staining were set at 10³ for PI staining. A figure exemplifying the gating strategy is provided in the Supplementary Fig. 1.

**Immunofluorescence (IF)**. Primary human hepatocytes were seeded on 8-well chamber slides (1.5 × 10⁴ cells/well) 24 h before the staining. Cells were fixed in 4% PFA for 20 min, washed with PBS, and non-specific sites were blocked with 5% BSA in PBS for 2 h. Cells were incubated with IL11RA, IL6R, gp130, or Albumin antibody overnight (4 °C), followed by incubation with the appropriate Alexa Fluor 488 secondary antibody for 1 h (RT). Negative control cells (−) were only stained with the secondary antibody. Chamber slides were dried in the dark and 5 drops of mounting medium with DAPI were added to the slides for 15 min prior to imaging by fluorescence microscope (Leica).

**Operetta high throughput phenotyping assay**. HSCs were seeded in 96-well black CellCarrier plates (PerkinElmer) at a density of 5 × 10³ cells per well. Following simulations, cells were fixed in 4% PFA (Thermo Fisher), permeabilized with 0.1% Triton X-100 (Sigma), and non-specific sites were blocked with 0.5% BSA and 0.1% Tween-20 in PBS. Cells were incubated overnight (4 °C) with primary antibodies (1:500), followed by incubation with the appropriate Alexa Fluor 488 secondary antibodies (1:1000). Cells were counterstained with 1 µg/ml DAPI (D1306, Thermo Fisher in blocking solution. Each condition was imaged from duplicated wells and a minimum of 7 fields/well using Operetta high-content imaging system 1483 (PerkinElmer). Cells expressing ACTA2 were quantified using Harmony v3.5.2 (PerkinElmer) and the percentage of activated fibroblasts/total cell number (ACTA2[+ve]) was determined for each field. The measurement of fluorescence intensity per area (normalized to the number of cells) of Collagen I was performed with Columbus 2.7.1 (PerkinElmer).

**Oil Red O staining**. Primary human hepatocytes were seeded on 8-well chamber slides (1.5 × 10⁴ cells/well) Following 24 h of palmitate treatment, cells were fixed in 10% PFA for 30 min, washed with distilled water, and incubated with 60% (v/v) isopropyl alcohol for 5 min. Cells were then stained with Oil Red O Solution (O0625, Sigma) for 30 min and washed with distilled water prior to imaging with a bright field microscope (BX53, Olympus). The lipid droplets were identified by their red staining.

**Reactive oxygen species (ROS) detection**. Primary human hepatocytes were seeded on 8-well chamber slides (1.5 × 10⁴ cells/well). For this experiment, cells were not serum-starved prior to palmitate treatment. Twenty-four hours following palmitate stimulation, cells were washed, incubated with 25 µM of DCFDA solution (ab113851, Abcam) for 45 min at 37 °C in the dark, and rinsed with the dilution buffer according to the manufacturer's protocol. Live cells with positive DCF staining were imaged with a filter set appropriate for fluorescein (FITC) using a fluorescence microscope (Leica).

**Seahorse assay**. Primary human hepatocytes were seeded into the Seahorse XF Cell Culture Microplate (1 × 10⁴ cells/well) and serum-starved overnight prior to stimulations. Seahorse measurements were performed on Seahorse XFe96 Extracellular Flux analyzer (Agilent). XF Cell Mito Stress Test kit (103015-100, Agilent) was used to measure the mitochondrial oxygen consumption rate as per the manufacturer's protocol. Briefly, stimulation media were removed and replaced with 180 µl of Mitostress assay medium at 37 °C. Oligomycin (1 µM; ATP synthase inhibitor) was injected following basal OCR measurements followed by injection of FCCP (1 µM; an uncoupling agent that collapses the proton gradient and disrupts the mitochondrial membrane potential), and finally by injection of a mixture of rotenone (1 µM; a complex I inhibitor) and antimycin A (1 µM; a complex III inhibitor). The percentage of fatty acid oxidation analysis was performed by using Seahorse XF Mito Fuel Flex Test kit (103260-100). Acute injections of CPT1 alpha inhibitor Etomoxir (4 µM) was used to inhibit mitochondrial FAO, whereas BPTES (3 µM) and UK5099 (2 µM) were used to inhibit mitochondrial glutamine and glucose oxidation, respectively, to inhibit 100% mitochondrial fuel oxidation. Seahorse Wave Desktop software was used for report generation and data analysis.

**RNA-sequencing (RNA-seq) and ribosome profiling (Ribo-seq)**
*Generation of RNA-seq libraries.* Total RNA was extracted from human hepatocytes using RNeasy columns (Qiagen). RNA was quantified using a Qubit RNA High-Sensitivity Assay kit (Life Technologies) and its quality was assessed on the basis of their RNA integrity number using the LabChip GX RNA Assay Reagent Kit (PerkinElmer). TruSeq Stranded mRNA Library Preparation kit (Illumina) was used to measure transcript abundance following standard instructions from the manufacturer.

**Generation of Ribo-seq libraries**. Hepatocytes were grown to 90% confluence in a 10 cm culture dish and lysed in 1 ml cold lysis buffer (formulation as in TruSeq® Ribo Profile Mammalian Kit, RPHMR12126, Illumina) supplemented with 0.1 mg/ml cycloheximide. Homogenized and cleared lysates were then footprinted with Truseq Nuclease (Illumina) according to the manufacturer's instructions. Ribosomes were purified using Illustra Sephacryl S400 columns (GE Healthcare), and the protected RNA fragments were extracted with a standard phenol:chloroform: isoamylalcohol technique. Following ribosomal RNA removal (Mammalian Ribo-Zero Magnetic Gold, Illumina), sequencing libraries were then prepared out of the footprinted RNA by using TruSeq® Ribo Profile Mammalian Kit according to the manufacturer's protocol.

The final RNA-seq and ribosome profiling libraries were quantified using KAPA library quantification kits (KAPA Biosystems) on a StepOnePlus Real-Time PCR system (Applied Biosystems) according to the manufacturer's protocol. The quality and average fragment size of the final libraries were determined using a LabChip GX DNA High Sensitivity Reagent Kit (PerkinElmer). Libraries with unique indexes were pooled and sequenced on a NextSeq 500 benchtop sequencer (Illumina) using NextSeq 500 High Output v2 kit and single-end 75-bp sequencing chemistry.

**Data processing and analyses for RNA-sequencing and ribosome profiling**. Raw sequencing data were demultiplexed with bcl2fastq *V2.19.0.316* and the adaptors were trimmed using *Trimmomatic*[36] *V0.36*, retaining reads longer than 20 nt post-clipping. Ribo-seq reads were aligned using bowtie[37] to known mtRNA, rRNA, and tRNA sequences (RNACentral[38], release 5.0) and only unaligned reads were retained as Ribosome protected fragments (RPFs). Alignment to the human genome (hg38) was carried out using STAR[39]. Gene expression was quantified on the CDS (coding sequence) regions for Ribo-seq and exonic regions for RNA-seq using uniquely mapped reads (Ensembl database release GRCh38 v86) with feature counts[40]. TPM was calculated and visualized using boxplot to compare baseline expression of IL11RA (ENSG00000137070), IL6R (ENSG00000160712), and gp130 (ENSG00000134352). Read coverage using Ribo-seq and RNA-seq reads for IL11RA, IL6R, and gp130 was visualized using Gviz R package[41] with strand-specific alignment files.

**Animal models**. Mice were housed in temperatures of 21–24 °C with 40–70% humidity on a 12 h light/12 h dark cycle and provided with food and water ad libitum.

**Mouse models of metabolic liver disease**.

1. HFMCD
   6–8-week-old C57BL/6N, *Il11ra1*$^{-/-}$ mice, and *Il11ra1*$^{loxP/loxP}$ and their respective control were fed with methionine- and choline-deficient diet supplemented with 60 kcal% fat (HFMCD, A06071301B, Research Diets) for 4 weeks. Control mice received normal chow (NC, Specialty Feeds).
2. WDF
   6–8-week-old C57BL/6N, *Il11ra1*$^{-/-}$ mice, and *Il11ra1*$^{loxP/loxP}$ and their respective control were fed western diet (D12079B, Research Diets) supplemented with 15% weight/volume fructose in drinking water (WDF) for 16 weeks. Control mice received NC and tap water.

***Il11ra1*-deleted mice (KO)**. Six- to eight-week-old male *Il11ra1*$^{-/-}$ mice (B6.129S1-*Il11ra*$^{tm1Wehi}$/J, Jackson's Laboratory) were intravenously injected with $4 \times 10^{11}$ genome copies (gc) of AAV8-*Alb-mbIl11ra1* or AAV8-*Alb-sIl11ra1* virus to induce hepatocyte-specific expression of mouse *Il11ra1* or soluble *Il11ra1*, respectively. As controls, both *Il11ra1*$^{-/-}$ mice and their wild-type littermates (*Il11ra1*$^{+/+}$) were intravenously injected with $4 \times 10^{11}$ gc AAV8-Alb-Null virus. Three weeks after virus injection, mice were fed with HFMCD, WDF, or NC. Durations of diet are outlined in the main text and/or figure legends.

**In vivo administration of soluble gp130**. Six- to eight-week-old male C57BL/6N mice (InVivos) were injected with $4 \times 10^{11}$ gc AAV8-Alb-*sgp130* virus to induce hepatocyte-specific expression of soluble gp130; control mice were injected with $4 \times 10^{11}$ gc AAV8-Alb-Null virus. Three weeks following virus administration, mice were fed with HFMCD, WDF, or NC for durations that are outlined in the main text and/or figure legends.

***Il11ra*-floxed mice (CKO)**. *Il11ra*-floxed mice, in which exons 4–7 of the *Il11ra1* gene were flanked by loxP sites, were created using the CRISPR/Cas9 system as previously described[42]. To induce the specific deletion of *Il11ra1* in hepatocytes, 6–8-week-old male homozygous *Il11ra1*-floxed mice were intravenously injected with AAV8-Alb-Cre virus ($4 \times 10^{11}$ gc); a similar amount of AAV8-Alb-Null virus were injected into homozygous *Il11ra1*-floxed mice as controls. The AAV8-injected mice were allowed to recover for 3 weeks prior to HFMCD, WDF, or NC feeding. Knockdown efficiency was determined by western blotting of hepatic IL11RA.

**RT-qPCR**. Total RNA was extracted from snap-frozen liver tissues using Trizol (Invitrogen) and RNeasy Mini Kit (Qiagen). PCR amplifications were performed using iScript cDNA Synthesis Kit (Bio-Rad). Gene expression was analyzed in duplicate by SYBR green (Qiagen) technology using StepOnePlus (Applied Biosystems) over 40 cycles. Expression data were normalized to *GAPDH* mRNA expression and fold change was calculated using $2^{-\Delta\Delta Ct}$ method. The primer sequences are listed in Supplementary Table 1.

**Immunoblotting**. Western blots were carried out on total protein extracts from hepatocytes and liver tissues. Hepatocyte and liver tissue lysates were homogenized in RIPA Lysis and Extraction Buffer (89901, Thermo Scientific) containing protease and phosphatase inhibitors (Roche). Protein lysates were separated by SDS-PAGE and transferred to PVDF membranes. Protein bands were visualized using the ECL detection system (Pierce) with the appropriate secondary antibodies: anti-rabbit HRP, anti-mouse HRP, or anti-rat HRP. Uncropped western blot images are provided in Source data file.

**Colorimetric assays**. Alanine aminotransferase (ALT) activity in the cell culture supernatant and mouse serum was measured using ALT Activity Assay Kit (ab105134, Abcam). Cellular and liver glutathione (GSH) levels were measured using Glutathione Colorimetric Detection Kit (EIAGSHC, Thermo Fisher). Total hydroxyproline content in mouse livers was measured using Quickzyme Total Collagen assay kit (QZBtotco15, Quickzyme Biosciences). The levels of triglycerides in hepatocyte lysates and in mouse serum and livers were measured using Triglyceride Assay Kit (ab65336, Abcam). Mouse serum levels of aspartate aminotransferase (AST), cholesterol, and β-hydroxybutyrate were measured using AST Assay Kit (ab105135, Abcam), Cholesterol Assay Kit (ab65390; Abcam), and beta-hydroxybutyrate (Ketone body) Colorimetric Assay Kit (700190; Cayman Chemicals), respectively. All colorimetric assays were performed according to the manufacturer's protocol.

**Enzyme-linked immunosorbent assay (ELISA)**. The levels of IL11, IL6, CCL2, and CCL5 in equal volumes of cell culture media were quantified using Human IL11 Quantikine ELISA kit (D1100; R&D Systems), Human IL-6 Quantikine ELISA Kit (D6050; R&D Systems), Human CCL2/MCP-1 Quantikine ELISA Kit

(DCP00; R&D Systems), and Human CCL5/RANTES Quantikine ELISA Kit (DRN00B; R&D Systems), respectively. The levels of IL11, IL6, gp130, and IL11RA in mouse serum were quantified using Mouse IL11 DuoSet ELISA (DY418; R&D Systems), Mouse IL6 Quantikine ELISA Kit (M6000B; R&D Systems), Mouse Interleukin 11 Receptor Alpha (IL11Ra) ELISA Kit (MBS452535; MyBioSource), and Mouse gp130 DuoSet ELISA (DY468, R&D Systems), respectively. All ELISA assays were performed according to the manufacturer's protocol.

**Liver tissue processing and histological analysis**.

1. Immunohistochemistry (IHC)
   Comparison of IL11RA and IL6R expression in healthy human liver (ab4348, Abcam) and in mouse liver: mouse livers were fixed in 10% neutral-buffered formalin (NBF), paraffinized, cut into 7-μm sections. Both human and mouse liver tissue sections were incubated with primary antibodies overnight and visualized using an ImmPRESS HRP anti-rabbit IgG polymer detection kit (MP-7401, Vector Laboratories) with ImmPACT DAB Peroxidase Substrate (SK-4105, Vector Laboratories).
   Comparison of IL11RA expression in the human livers from healthy control and patients suffering from NASH: these studies were outsourced to a company (Fibrofind). Briefly, paraffin embedded blocks of human livers were cut into slides and stained with IL11RA antibody (NBP2-32671, Novus Biologicals) or control. Analysis was performed on liver sections from three healthy individuals as control (NHL20, NHL60, NHL67; 1 section/code) and on liver sections from 2 NASH patients (TLPAT5, TLPAT14; 1 section/ code). Due to data protection policies, no further information is available on samples or patients.
2. H&E and Masson's Trichrome staining
   Mouse liver samples were processed and sectioned as mentioned above, followed by hematoxylin and eosin (H&E) or Masson's Trichrome staining according to standard protocol.

**Statistical analysis**. All statistical analyses were performed using GraphPad Prism software (version 6.07). Simple two-tailed Student's *t*-tests were used for experimental setups requiring testing of just two conditions. For comparisons between more than two conditions, one-way ANOVA with Dunnett's correction (when several conditions were compared to one condition) or Tukey's correction (when several conditions were compared to each other) were used. Comparisons of two parameters (body weight across time) for different groups were performed by two-way ANOVA with Tukey's correction. The criterion for statistical significance was set at $P < 0.05$.

**Reporting summary**. Further information on research design is available in the Nature Research Reporting Summary linked to this article.

## Data availability

All data are available within the Article or Supplementary Information. The RNA-seq and RIBO-seq data reported in this paper are available in NCBI BioProject ID: PRJNA670552. Source data are provided with this paper.

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

## Acknowledgements

This research is supported by the National Medical Research Council (NMRC), Singapore STaR awards (NMRC/STaR/0029/2017), NMRC Center Grant to the NHCS, MOH-CIRG18nov-0002, Goh Foundation, Tanoto Foundation. A.A.W. is supported by NMRC/OFYIRG/0053/2017. P.M.Y. is supported by NMRC/CIRG/1457/2016. The authors would like to acknowledge the technical support of B.L. George and S. Lim. We thank Fibrofind for helping with IL11RA staining in patients with NASH.

## Author contributions

S.A.C. and A.A.W. conceived and designed the study. J.D., S.V., E.A., B.N., W.W.L., B.K.S., J.T., M.W., and A.A.W. performed in vitro cell culture, cell biology, and molecular biology experiments. J.D., J.Z., M.T., S.G.S., and N.S.J.K. performed in vivo studies. S.G.S. and S.Y.L. performed histology analysis. J.D., S.P.C., P.M.Y., S.S., S.A.C., and A.A.W. analyzed the data. L.P.M. and E.A. performed RNA and Ribo sequencing. J.D., E.A., S.A.C., and A.A.W. prepared the manuscript with input from co-authors.

## Competing interests

S.A.C., S.S., A.A.W., B.N., W.W.L., and B.K.S. are co-inventors on a number of patent applications relating to the role of IL11 in human diseases that include the published patents: WO2017103108, WO2017103108 A2, WO 2018/109174 A2, WO 2018/109170 A2. S.A.C. and S.S. are co-founders and shareholders of Enleofen Bio PTE LTD, a company (which S.A.C. is a director of) that developed anti-IL11 therapeutics, which were acquired for further development by Boehringer Ingelheim. All other authors declare non-competing interests.
