## [Peer Review File · Nature Communications]

Reviewers' Comments:

Reviewer #1:

Remarks to the Author:

The manuscript by Dong et al investigates contribution of IL-11- and IL-6-induced signaling pathway in the development of non-alcoholic steatosis hepatitis using various murine models. The authors show that trans-signaling of IL-11 and IL-6 do not play a major role in the development of lipotoxicity of hepatocytes and subsequent liver injury. However, trans-signaling of IL-6 in the development of NASH has been published elsewhere (Kammoun et al, PLoS ONE 2017: <https://doi.org/10.1371/journal.pone.0179099>). In sharp contrast, IL-11/IL-11RA signal plays a crucial role in promoting JNK activation, NOX4-dependent ROS accumulation and subsequent hepatocyte apoptosis, resulting in liver injury. Although the results presented are potentially interesting, there are serious concerns about the authors' experimental conditions. Moreover, the mechanisms underlying IL-11-induced apoptosis are not fully investigated. The followings are specific comments.

Major points.

1. In Figure 1A, The results do not include negative control. The authors need to repeat the experiment using the liver of *Il1ra1*^{-/-} mice. Please indicate the length of the scale bars in Figure legend. Also, please add magnified views to show which types of cells are stained with anti-IL11RA antibody.
2. In Figure 1F, it is unclear why hyper IL-11 does not induced phosphorylation of STAT3. I think that IL-11 binds to IL-11Ra and induces signals via gp130. In addition, the authors need to repeat the same experiments using recombinant human IL-11 and IL-6.
3. In Figure 1G, H, in addition to the ratios of ALT before and after stimulation of hyper IL-11, the authors need to show the concentrations of ALT before and after stimulation.
4. In Figure 1L, the authors claim that IL-11 induces caspase 3-dependent cell death. To verify the authors' claim, the authors need to test whether IL-11-induced cell death is blocked in the presence of caspase inhibitors such as zVAD-fmk or qVD-OPH. Moreover, if the signaling pathways triggered by JNK or ERK contribute to cell death, the authors need to test whether inhibitors of JNK or ERK attenuate IL-11-induced cell death. Moreover, it is unclear NOX4 activation induces apoptosis. Thus, the authors need to investigate the mechanisms in more detail.
5. In Figure 2, the authors show the effectiveness of anti-human IL11RA antibody X209 against palmitate-induced lipotoxicity in primary human hepatocytes. To confirm that the protective effect of X209 is solely by blocking IL-11 cis-signaling and not by cross-reaction with other factors, the authors need to perform a similar experiment using mouse primary hepatocytes from *Il11ra1*^{-/-} mice used in Supplementary Figure 8. The hepatocytes from *Il11ra1*^{-/-} mice will be more resistant to palmitate-induced lipotoxicity than those from *Il11ra1*^{+/+} mice. The molecular mechanism of palmitate-induced hepatotoxicity is thought to be rather complex. Several reports suggest that palmitate treatment induces ROS generation via ER stress. In addition, palmitate is also reported to activate TLR4 to induce inflammatory response. To fully understand the mechanism of lipotoxicity, it will be informative to examine the markers of ER stress or TLR4 activation.
6. Figure 2E, again, the authors need to test whether caspase inhibitor blocks palmitate-induced apoptosis.
7. Figure 2H, assuming that ROS are responsible for induction of hepatocyte apoptosis, the authors need to check the effect of NOX inhibitor, DPI on IL-11-induced cell death.
8. In Figure 3A, the authors need to show how many percentages of hepatocytes are infected with AAV8 virus and how long infected viruses remain in hepatocytes after injection under the authors' experimental conditions.
9. In Figure 3D, it is unclear why injection of AAV8 encoding *sgp130* induces elevation of serum IL-6 (approximately 20 ng/ml!) without WDF. The authors need to explain the mechanism underlying elevation of IL-6. Related to this, serum concentrations of IL-6 show a twenty-fold high concentrations of IL-11 (20 ng/ml v.s. 1 ng/ml). However, the signaling intensities of Western blotting of IL-11 is higher than those of IL-6. Based on the Western blotting, IL-6 signals appear to be undetectable. The authors need to explain these apparent discrepancy.

10. In Figure 4, the authors test whether expression of IL-11RA on hepatocytes is responsible for exacerbation of NASH by injecting AAV8-Alb-cre into Il11ra1flox/flox mice. The reviewer has no idea why the authors do not generate and use hepatocyte-specific Il11ra1-deficient mice by crossing Il11ra1flox mice with Albmin-Cre Tg mice for these experiments. The authors need to repeat the same experiments using Il11raflox/flox;Alb-Cre mice. If there is discrepancy of the effect of deletion of Il11ra on the development of NASH in between AAV8-Alb-cre-injected Il11raflox/flox mice and Il11raflox/flox;Alb-Cre mice, the authors need to discuss this point. We cannot formally exclude the possibility that injection of large amounts of AAV8 virus along with deletion of Il11ra in hepatocytes might induce artifacts.

11. In Figure 6B, according to the experimental design described in Supplementary Figure 7A, sIL11-RA is ~60 amino acid shorter than mbIL11-RA. However, Western blot in Figure 6b shows that the electrophoretic mobility of sIL11RA is similar to those of native IL11-RA and mbIL11-RA. Please explain the reason of this discrepancy.

Minor points.

1. The authors need to describe the detailed conditions of their experiments, including how long cells were stimulated with the indicated agents, how many mice were used in each experiment, and how many times these experiments were repeated.
2. The authors need to include molecular size markers in all results of Western blotting.

Reviewer #2:

Remarks to the Author:

This manuscript describes the autocrine role of IL11 signaling in driving the progression of NASH. The main questions the authors focus on are: a) whether IL11 signals to hepatocytes through cis (membrane-bound receptor) or trans (soluble receptor) mechanisms; 2) the specific role of IL11 in lipotoxicity; and 3) whether cis- and trans- IL11 signaling have effects on NASH progression.

Some results in this manuscript are somewhat expected as the role of IL11 signaling in NASH has been extensively studied and recently published (Gastroenterology, 2019, PMID: 31078624). However, the authors used different approaches with a combination of in vitro and in vivo data, which provide great novelty to this manuscript. The writing and organization of the manuscript are satisfactory.

The major drawback of the manuscript is that it is unclear if the trans-signaling mechanism exists in the context of NASH and if it indeed signals to hepatocytes. Likewise, it is unclear if cis- and trans-signaling regulate the same or different downstream pathways.

Specific comments:

1. In Figures 1L and S2, the authors indicate that the addition of gp130 cannot block the effects from IL11 treatment. However, this may also occur because sIL11RA is not present in the culture media. This needs to be clarified.
2. In Figures 1L and S2, additional sIL11RA does not promote the effects. This may result from the cis signal not reaching the saturation point. Hence, the addition of sIL11RA is likely to not affect the level of IL11, and gp130 remains the same. Therefore, the results may only indicate that the trans- signal is equal to or weaker than the cis signal. It is recommended to isolate hepatocytes from IL11RA knockout mice to completely shut down the cis signaling in the hepatocytes. Then, treat the cells with IL11 in the presence of sIL11RA to observe if there is apoptosis through the ERK-JNK axis.
3. There is no evidence showing that sIL11RA is increased in human and mouse NASH. Hence, the rationale for studying the role of trans- signal should be solidified.

Minor comments:

1. Figure 1A, please enhance the magnification and show the IL11 expression pattern on the

plasma membrane of hepatocytes and IL6R on Kupffer cells.

2. Figure 1A, the size of the scale bar is missing.

3. The statement that "only a few hepatocytes expressed low amounts of IL6R" is not entirely correct. In Figure 1B, most of the cells stained by IL6R still have higher fluorescence intensity than the IgG control. This means, most of the cells are still expressing IL6R but at a much lower level than IL11RA and gp130.

4. Although it is in medium, the true value of ALT should be indicated in Figure 1 as in Figure 2.

5. Palmitate induces less than 1ng/ml of IL11. Based on the data in Figure 1 and S2, IL11 at this concentration did not induce a major difference in ALT (Figure S2B). Therefore, it is difficult to conclude that IL11-IL11RA signaling is a major contributor to palmitate-induced cell death.

6. Figure 2H, it is better to show reduced and total glutathione.

7. Figure 2I, DCFDA should be stained together with DAPI to show the same cell density.

8. Figure 2K, increase the magnification of the Oil Red O staining as it is difficult to assess the changes in fatty acid uptake. It is also recommended to measure intracellular TG and NEFA (maybe normalized by protein concentration) to strengthen the conclusion.

9. It is not clear why FASN is mentioned in the results. This is a protein involved in lipogenesis. However, it is not commented anywhere else.

10. Figure 3, again, there is no evidence showing that sIL11RA is present in the environment.

11. Figure 4C, absolute b.w. number is recommended, and the NC WT curve is missing.

12. Figure 4D, not clear the H&E is from which time point.

Reviewer #3:

Remarks to the Author:

The study by Dong et al. aimed to examine a role for IL11 signaling in hepatocyte lipotoxicity in relation to the pathogenesis of non-alcoholic steatohepatitis (NASH). For this purpose, the authors performed animal studies involving various NASH models and showed that hepatocyte-specific deletion of IL11ra1 protects mice from all aspects of NASH. The authors also showed that restoration of IL11 cis-signaling in hepatocytes only in mice globally deleted for IL11ra1 reconstitutes steatosis and inflammation. Based on their results, the authors conclude that autocrine IL11-mediated cell death underlies hepatocyte lipotoxicity and that liver fibrosis and inflammation occur subsequently. Overall, this is an interesting study. However, the study overly stated the role of autocrine IL11 activity in hepatocytes in promoting steatohepatitis and nearly ignored completely the roles of factors derived from IL11-driven hepato-lipotoxicity on activating liver Kupffer cells/macrophages and hepatic stellate cells.

Specific comments

1. This manuscript has included strong data to support a detrimental role for IL11 cis-signaling in hepatocytes in promoting NASH.

2. Since IL11 cis-signaling is critical to hepatocyte lipotoxicity, it is important to show whether IL11RA expression in liver sections from NASH patients differs from that in normal liver sections.

3. Data in Fig. 1L,K and Fig. 2G,H,J indicate a predominant role for IL11 cis-signaling in causing hepatocyte death.

4. Data in Fig. 3B is quite interesting. However, this reviewer is curious if IL11RA is differentially expressed in liver sections from WDF-Null mice vs. that from WDF-sgp130 mice. It is also important to demonstrate the expression pattern of IL11RA in hepatocytes and non-parenchymal cells (NPCs) from liver sections; given the particular importance of NPCs in promoting NASH in response to hepatocyte-derived factors.

5. While data in Fig. 3O,P indicated increased liver inflammation in either WDF-Null mice or WDF-sgp130 mice vs. NC-Null mice, it is not clear about the proportional contribution of increased inflammatory responses in hepatocytes vs. NPCs to liver inflammation.

6. Fig. 4C,D: HFMCD-fed CKO mice displayed significantly decreased hepatic steatosis while showing no decrease in body weight. In contrast, HFMCD-WT mice displayed massive hepatic steatosis while also showing a marked decrease in body weight. This reviewer is curious about the

mechanisms by which hepatocyte-specific IL11RA disruption decreases hepatic steatosis. This reviewer is also curious if increased adipose tissue lipolysis and fat flow to the liver in WT mice contributed to hepatic steatosis and whether this mechanism was impaired in HFMCD-CKO mice.

7. For mouse models in Fig. 4, what were the serum levels of IL11?

8. For data in Fig. 4 and Fig 5: it appears that upon HFMCD feeding CKO mice recovered from weight loss or gained body weight compared with WT mice. However, upon WDF feeding, CKO mice revealed a smaller gain in body weight compared with WT mice. Are there any explanations for why CKO mice responded differently to different diets? This is important because adiposity or fat mass is a key factor determining hepatic steatosis.

9. For Fig. 5E: what are mechanisms for decreased hepatic steatosis in WDF-CKO mice?

10. Fig. 6: the data are strong in terms of validating a role for activation of IL11 cis-signaling in promoting NASH.

11. Fig. 7: the paracrine actions of hepatocytes on HSCs have not been validated. Indeed, hepatocyte factors, generated in response to activation of IL11 cis-signaling, could act on HSCs and accounts for, in large part, the fibrogenic activation of HSCs.

12. Similar to that described in Point 11, the paracrine actions of hepatocyte-derived factors on liver Kupffer cells/macrophages could account for, in large part, liver inflammation. This point has not even been mentioned.

Point-by-point responses to the comments made by Reviewers at Nature Communications

Reviewer #1 (Remarks to the Author):

The manuscript by Dong et al investigates contribution of IL-11- and IL-6-induced signaling pathway in the development of non-alcoholic steatohepatitis using various murine models. The authors show that trans-signaling of IL-11 and IL-6 do not play a major role in the development of lipotoxicity of hepatocytes and subsequent liver injury. However, trans-signaling of IL-6 in the development of NASH has been published elsewhere (Kammoun et al, PLoS ONE 2017: <https://doi.org/10.1371/journal.pone.0179099>).

Author response: We are pleased that the Reviewer believes we have shown that neither IL11 nor IL6 *trans*-signaling play a major role in the development of hepatic lipotoxicity. We would like to point out that the above-mentioned paper concluded that a role for IL6 [and IL11] *trans*-signaling could not be excluded, based on the model used. In our manuscript, we discount a role for both IL6 or IL11 *trans*-signaling *in vitro* and in two independent mouse NASH models, which is complemented by *in vitro* data.

In sharp contrast, IL-11/IL-11RA signal plays a crucial role in promoting JNK activation, NOX4-dependent ROS accumulation and subsequent hepatocyte apoptosis, resulting in liver injury. Although the results presented are potentially interesting, there are serious concerns about the authors' experimental conditions. Moreover, the mechanisms underlying IL-11-induced apoptosis are not fully investigated. The followings are specific comments.

Major points.

1. In Figure 1A, The results do not include negative control. The authors need to repeat the experiment using the liver of Il1ra1^{-/-} mice. Please indicate the length of the scale bars in Figure legend. Also, please add magnified views to show which types of cells are stained with anti-IL11RA antibody.

Author response: While it is routine to provide a negative control without the use of primary antibody, which the reviewer rightly points out we omitted in the earlier version of the manuscript (for space reasons), it is not usually expected to test antibodies against a null background, as reagents (e.g. knockout mice) are often lacking. However, for completeness, and at the request of the reviewer, we now provide IL11RA staining on wild-type and *Il1ra1^{-/-}* mouse liver for antibody validation purposes as well as the conventional negative control (revised **Supplementary Fig. 2A**). We also provide magnified views of **Fig. 1A** in the revised manuscript.

2. In Figure 1F, it is unclear why hyper IL-11 does not induced phosphorylation of STAT3. I think that IL-11 binds to IL-11Ra and induces signals via gp130. In addition, the authors need to repeat the same experiments using recombinant human IL-11 and IL-6.

Author response: The Reviewer highlights a key aspect of IL11 signaling that we show experimentally here in hepatocytes (as we have previously in fibroblasts ¹) and which we discussed. It is expected that neither hyperIL11 nor IL11 will readily activate STAT3 when given at physiologically relevant doses as IL11 itself or IL11:IL11RA complexes bind to gp130 and preferentially activate non-canonical ERK signaling (not STAT). In the manuscript we refer to a reference that goes into this matter in detail ². We also stated in the results section:

“HyperIL11, like IL11 itself, dose-dependently activated ERK and JNK. Similarly, IL6 *trans*-signaling dose-dependently induced STAT3 phosphorylation, as seen with IL6 itself, but did not activate ERK or JNK (**Fig. 1F**). Thus, IL11 or IL6 (*cis* and *trans*) signaling results in activation of different intracellular pathways in hepatocytes, which is a novel finding.”

To look into these signaling aspects in even greater detail we have performed additional experiments, as requested by the Reviewer, and stimulated hepatocytes with IL11 or IL6. As expected, we observe the same pattern of activation of ERK by IL11 and STAT3 by IL6, as shown below. We have added Western blots showing ERK and STAT3 activation status following IL11 and IL6 stimulation as **Supplementary Fig. 4A**.

Figure 1F

Supplementary Figure 4A

3. In Figure 1G, H, in addition to the ratios of ALT before and after stimulation of hyper IL-11, the authors need to show the concentrations of ALT before and after stimulation.

Author response: At the Reviewer’s request, we now show ALT concentration (U/L) throughout the manuscript.

4. In Figure 1L, the authors claim that IL-11 induces caspase 3-dependent cell death. To verify the authors’ claim, the authors need to test whether IL-11-induced cell death is blocked in the presence of caspase inhibitors such as zVAD-fmk or qVD-OPH. Moreover, if the signaling pathways triggered by JNK or ERK contribute to cell death, the authors need to test whether inhibitors of JNK or ERK attenuate IL-11-induced cell death. Moreover, it is unclear NOX4 activation induces apoptosis. Thus, the authors need to investigate the mechanisms in more detail.

Author response: The Reviewer highlights an important aspect, which is complex to dissect fully and that we are careful not to over-interpret given the cross-talk between signaling pathways in NASH pathogenesis. Hepatocyte death in NASH is also complex and thought to involve variable contributions of apoptosis, necrosis, pyroptosis, necroptosis and other forms of cell death.

We recently found that NOX4 upregulation in hepatocytes is upstream of both ERK/JNK activation and caspase activation³. It is possible that NOX4-derived ROS also impinge on additional, yet to be discovered pathways, and we do not propose to use pharmacological inhibition of ERK/JNK or other pathways. Instead, we focused additional mechanistic studies on the two nodal points of the proposed pathway: (1) NOX4 activation, which we hypothesized is an upstream event and (2) caspase activation, likely downstream, and now present these data as **Supplementary Fig. 4E-H** and shown below.

Supplementary Figure 4 E-H

We interpret these new data and discuss in greater detail the complexities of signaling and cell death in the revised manuscript as shown here:

Revised Manuscript Results:

“NOX4 inhibitors reduced IL11-induced ERK and JNK activation and robustly protected hepatocytes from IL11-induced cell death (**Supplementary Fig. 4E and F**). Pan-caspase inhibition, while protective, was not as effective as NOX4 inhibition in preventing cell death and did not reduce either NOX4 induction or ERK activation (**Supplementary Fig. 4G and H**). This places NOX4 activation upstream of both ERK and caspase-3 activation in IL11-stimulated hepatocytes and suggests that apoptotic cell death is only one mode of cell death in this context.”

Revised Manuscript Discussion:

“The effects on ROS are likely complex as IL11 induces NOX4 but also impacts mitochondria function and perhaps the endoplasmic reticulum, directly or indirectly. Interestingly, inhibition of

NOX4 was more effective than caspase inhibition in reducing cell death and thus NOX4-induced ROS production lies upstream of ERK and caspase activation.”

5. In Figure 2, the authors show the effectiveness of anti-human IL11RA antibody X209 against palmitate-induced lipotoxicity in primary human hepatocytes. To confirm that the protective effect of X209 is solely by blocking IL-11 cis-signaling and not by cross-reaction with other factors, the authors need to perform a similar experiment using mouse primary hepatocytes from *Il11ra1*^{-/-} mice used in Supplementary Figure 8. The hepatocytes from *Il11ra1*^{-/-} mice will be more resistant to palmitate-induced lipotoxicity than those from *Il11ra1*^{+/+} mice.

Author response: We thank the Reviewer for this comment but believe that this additional experiment is not needed and may indeed give misleading results. The data shown in Figure 2 is generated in primary human hepatocytes that express IL11RA throughout development. Hepatocytes from the *Il11ra1*-deleted mouse may have adapted to *Il11ra1* deletion and we have ourselves documented differences between global, germline deletion of *Il11ra1* as compared to acute inhibition in adults using temporal gene deletion or antibody-based inhibition. By way of example, global germline deletion of *Il11ra1* paradoxically sensitises mouse hepatocytes to acetaminophen-induced hepatitis⁴ whereas deletion of *Il11ra1* in adults in hepatocytes only strongly inhibits acetaminophen-induced liver damage³.

Thus we used an alternative approach: we inhibited IL11 signaling in hepatocytes using an IL11 mouse monoclonal antibody (X203)^{5,6} and show this has the same cytoprotective effects as X209 (an IL11RA mouse monoclonal antibody) (**Rebuttal Fig. 1.1**). This provides orthogonal validation [and specificity] for the importance of IL11 signaling in palmitate-induced cell death.

Rebuttal Fig.1.1 Effect of IL11 inhibition by X203 (IL11 antibody) on palmitate-induced ALT release (n=3).

The molecular mechanism of palmitate-induced hepatotoxicity is thought to be rather complex. Several reports suggest that palmitate treatment induces ROS generation via ER stress. In addition, palmitate is also reported to activate TLR4 to induce inflammatory response. To fully understand the mechanism of lipotoxicity, it will be informative to examine the markers of ER stress or TLR4 activation.

Author response: We agree with the Reviewer that this is a complicated matter and it is something we intend to dissect in detail over the coming years. As suggested, we provide new

information on the status of markers of ER stress here (**Rebuttal Fig. 1.2**). As shown below, ER stress marker proteins (CHOP and XBP1-s) were strongly induced by palmitate treatment whereas inhibition of IL11 signaling by X209 robustly inhibited their upregulation. Thus, we are in an agreement with the Reviewer, and believe that lipotoxicity by palmitate is mediated by complex factors including TLR4 activation, ROS/Oxidative stress, and ER stress among others pathways. This results in metabolic dysfunction and more than one form of cell death. Inhibiting IL11 signaling protects against palmitate-induced lipotoxicity as evident by inhibited NOX4 and ER stress markers as well as caspase 3 activation. We have not added these ER-related data to the manuscript so as not to over-complicate matters, but we do refer to ER-related effects in the revised discussion.

Rebuttal Figure 1.2. Western blot of palmitate-treated primary human hepatocytes in the presence of either IgG, X209 or sgp130. ER stress markers (CHOP and XBP1-s) are highlighted in red).

6. Figure 2E, again, the authors need to test whether caspase inhibitor blocks palmitate-induced apoptosis.

Author response: As suggested by the Reviewer, we have now provided data on the effect of pan caspase inhibitor (Z-VAD-FMK) as **Supplementary Fig. 5C-D** and show a partial dependency of palmitate-induced hepatocyte cell death on caspase 3 activation. We highlight that additional complexity is at play and other forms of cell death are involved, which we discuss in the revision.

Supplementary Figure 5 C-D

7. Figure 2H, assuming that ROS are responsible for induction of hepatocyte apoptosis, the authors need to check the effect of NOX inhibitor, DPI on IL-11-induced cell death.

Author response: We refer the Reviewer to our response for Question 4. As mentioned above, we have now provided these data as **Supplementary Fig. 4E-F** in the revised manuscript.

8. In Figure 3A, the authors need to show how many percentages of hepatocytes are infected with AAV8 virus and how long infected viruses remain in hepatocytes after injection under the authors' experimental conditions.

Author response: The use of adenovirus AAV8 with an albumin promoter is an established technique to manipulate gene expression in the liver⁷. In this model, it is accepted that the most important functional aspect to document is the effect of viral-mediated gene delivery on the protein levels of the gene of interest. This depends on composite effects relating to viral load, transduction efficiency, transgene mRNA expression, transgene protein expression, transgene immunogenicity and host response. In our case when we over-express or knockdown IL11RA or sgp130 we document effects at the protein level (locally and, where appropriate, systemically) as shown throughout the manuscript.

9. In Figure 3D, it is unclear why injection of AAV8 encoding sgp130 induces elevation of serum IL-6 (approximately 20 ng/ml!) without WDF. The authors need to explain the mechanism underlying elevation of IL-6. Related to this, serum concentrations of IL-6 show a twenty-fold high concentrations of IL-11 (20 ng/ml v.s. 1 ng/ml). However, the signaling intensities of Western blotting of IL-11 is higher than those of IL-6. Based on the Western blotting, IL-6 signals appear to be undetectable. The authors need to explain these apparent discrepancy.

Author response: This was a typographical error. The IL6 serum levels are now correctly reported as ~20 **pg/ml** (not ng/mL).

10. In Figure 4, the authors test whether expression of IL-11RA on hepatocytes is responsible for exacerbation of NASH by injecting AAV8-Alb-cre into *Il11ra1flox/flox* mice. The reviewer has no idea why the authors do not generate and use hepatocyte-specific *Il11ra1*-deficient mice by crossing *Il11ra1flox* mice with *Albmin-Cre Tg* mice for these experiments. The authors need to repeat the same experiments using *Il11raflox/flox;Alb-Cre* mice. If there is discrepancy of the effect of deletion of *Il11ra* on the development of NASH in between AAV8-Alb-cre-injected

Il11raflox/flox mice and Il11raflox/flox;Alb-Cre mice, the authors need to discuss this point. We cannot formally exclude the possibility that injection of large amounts of AAV8 virus along with deletion of Il11ra in hepatocytes might induce artifacts.

Author response: There are a number of reasons for our specific choice to use adenoviral gene delivery to increase / decrease IL11RA1 isoforms and/or sgp130 expression.

1. As referred to above (our response to Question 5), lifelong deletion of *Il11ra1* in hepatocytes (as would be achieved using albumin-Cre) results in hitherto uncharacterised adaptive responses in the liver that confound the use of this model in experiments. This effect can be large as reported previously where lifelong deletion of *Il11ra1* sensitises mice to acetaminophen injury ⁴, which is the opposite of what happens with *Il11ra1* inhibition in the adult using antibodies or temporally-regulated gene deletion.
2. In addition to deleting IL11RA for loss-of-function (LOF) experiments (**Fig. 4 and 5; Supplementary Fig. 8 and 9**), we studied (and compared) the effects of IL11RA1 gain-of-function (GOF) on an IL11RA1 null background in order to reconstitute IL11 *cis*-signaling in hepatocytes and also study *trans*-signaling by expressing soluble IL11RA1 (**Fig. 6; Supplementary Fig. 10 and 11**). By using AAV8-mediated delivery we can directly perform both LOF and GOF studies along with studying the *in vivo* GOF effects of sgp130 in WT mice on two NASH diets.
3. These various LOF and GOF experiments on WT or IL11RA1 null backgrounds could not have been performed using germline-based mouse transgenic models without the use of triple transgenic methodologies: IL11RA1^{-/-} x albumin-Cre x sIL11RA:flox-stop and IL11RA1^{-/-} x albumin-Cre x mL11RA:flox-stop. Furthermore, prior to contemplating these triple transgenic approaches transgenic mice for sIL11RA:flox-stop and mL11RA:flox-stop would need to be made as these mouse strains do not currently exist.
4. The Reviewer's comment that "*injection of large amounts of AAV8 virus along with deletion of Il11ra in hepatocytes might induce artifacts*". We point out that this is addressed by the use of the same doses of AAV8 null controls in all experiments throughout the manuscript.

11. In Figure 6B, according to the experimental design described in Supplementary Figure 7A, sIL11-RA is ~60 amino acid shorter than mbIL11-RA. However, Western blot in Figure 6b shows that the electrophoretic mobility of sIL11RA is similar to those of native IL11-RA and mbIL11-RA. Please explain the reason of this discrepancy.

Author response: Mouse IL11RA1 is composed of 432 amino acids with a theoretical molecular weight of 47 kDa. We are unsure as to why the ~6kDa difference between the isoforms is not apparent on the Western blot but gel resolving resolution or post-translational modifications could account for this. The specificity of the antibody for mouse IL11RA1 is assured as seen from the Western blot shown below for the two anti-IL11RA antibodies used in this study:

Rebuttal Figure 1.3. Western blots of IL11RA and GAPDH from *Il11ra*^{+/+} and *Il11ra*^{-/-} mouse atrial fibroblast lysate (n=4). IL11RA protein was probed using in-house (X209) or commercial (sc-130920, SantaCruz) mouse monoclonal IL11RA antibodies.

A clear difference in the membrane-bound IL11RA1 (mbIL11RA) vs the soluble IL11RA1 (sIL11RA) was apparent when we measured circulating levels of IL11RA, which are markedly elevated in the sIL11RA expressing mice (5.8-10 ng/ml) but not mbIL11RA expressing mice (0.26-0.57ng/ml) as shown in **Supplementary Fig 10B and 11C**, for HFMCD and WDF model, respectively (see below).

Minor points.

1. The authors need to describe the detailed conditions of their experiments, including how long cells were stimulated with the indicated agents, how many mice were used in each experiment, and how many times these experiments were repeated.

Author response: We have added these details in the figure legends and/or as part of the figure.

2. The authors need to include molecular size markers in all results of Western blotting.

Author response: Molecular size markers have been provided for all Western blot data.

Reviewer #2 (Remarks to the Author):

This manuscript describes the autocrine role of IL11 signaling in driving the progression of NASH. The main questions the authors focus on are: a) whether IL11 signals to hepatocytes through cis (membrane-bound receptor) or trans (soluble receptor) mechanisms; 2) the specific role of IL11 in lipotoxicity; and 3) whether cis- and trans- IL11 signaling have effects on NASH progression.

Some results in this manuscript are somewhat expected as the role of IL11 signaling in NASH has been extensively studied and recently published (Gastroenterology, 2019, PMID: 31078624). However, the authors used different approaches with a combination of in vitro and in vivo data, which provide great novelty to this manuscript. The writing and organization of the manuscript are satisfactory.

The major drawback of the manuscript is that it is unclear if the trans-signaling mechanism exists in the context of NASH and if it indeed signals to hepatocytes. Likewise, it is unclear if cis- and trans-signaling regulate the same or different downstream pathways.

Specific comments:

1. In Figures 1L and S2, the authors indicate that the addition of sgp130 cannot block the effects from IL11 treatment. However, this may also occur because sIL11RA is not present in the culture media. This needs to be clarified.

Author response: We agree that it is possible that the reason why sgp130 has no effect on potential IL11-related *trans*-signaling is that there is not sufficient sIL11RA present in the media. However, during revision we have checked levels and found approximately 250 pg/ml or sIL11RA in the media (**Rebuttal Fig. 2.1**). This clarifies for the Reviewer that sIL11RA is present in the media and *trans*-signaling could occur. This aside, we also added increasing amounts of IL11 to media pre-loaded with sIL11RA (1 µg/ml) or increasing amounts of sIL11RA to media pre-loaded with IL11. Reciprocal experiments were performed with sgp130. In no instance was there evidence of *trans*-IL11 signaling using these combined gain-of-function (GOF) and loss-of-function (LOF) approaches. However, in **Fig. 1H and I**, it is shown that pre-formed, man-made, synthetic HyperIL11 constructs have an effect and that this can be blocked with sgp130. These data suggest that a physiological role for IL11 *trans*-signaling in hepatocyte biology is unlikely, especially when considered with the *in vivo* data presented.

Rebuttal Figure 2.1. IL11RA levels in the hepatocyte media at baseline (n=3) as measured by Human IL11RA ELISA (MBS453817, MyBioSource).

2. In Figures 1L and S2, additional sIL11RA does not promote the effects. This may result from the *cis* signal not reaching the saturation point. Hence, the addition of sIL11RA is likely to not affect the level of IL11, and gp130 remains the same. Therefore, the results may only indicate that the *trans*- signal is equal to or weaker than the *cis* signal. It is recommended to isolate hepatocytes from IL11RA knockout mice to completely shut down the *cis* signaling in the hepatocytes. Then, treat the cells with IL11 in the presence of sIL11RA to observe if there is apoptosis through the ERK-JNK axis.

Author response: We refer the Reviewer to the discussion above and highlight that we performed a dose response for IL11. At the midpoint of the dose-response (~1.25 to 2.5 ng/ml IL11), the addition of 1 µg/ml of sIL11RA had no effect. Given that sIL11RA levels *in vivo* are orders of magnitude lower than the concentration we used we can conclude that if *trans*-signaling were to exist then its effects are negligible as compared to IL11 *cis*-signaling. This is also apparent from the *in vivo* experiments where loss of *cis*-signaling abrogates lipotoxicity but inhibition of *trans*-signaling with very high levels of sgp130 secreted from hepatocytes has no measurable effects in two independent NASH models. It may be possible to force an effect *in vitro* by using hepatocytes that are deleted for *Il11ra1* but based on *in vivo* data, this would most likely not be physiologically relevant.

3. There is no evidence showing that sIL11RA is increased in human and mouse NASH. Hence, the rationale for studying the role of *trans*- signal should be solidified.

Author response: In human and mouse NASH, IL11 levels are elevated⁵ and it is known that sIL11RA is present in human serum at levels ranging from 20 pg/ml - 4ng/ml⁸. Even though we do not know if sIL11RA is increased in patients with NASH, this does not matter as IL11 is largely upregulated and hence could form *trans*-complexes with the existing sIL11RA already present (i.e. sIL11RA levels themselves do not need to vary). Furthermore, we have now checked and documented elevated levels of sIL11RA in the serum of mice fed with Western diet and liquid fructose (WDF) (**Rebuttal Fig 2.2**). Thus it is apparent that IL11 *trans*-signaling could play a role in liver dysfunction and this has been postulated by others⁹.

Rebuttal Figure 2.2. Serum IL11RA levels in mice fed with normal chow (NC) or Western diet and 15% liquid fructose (n=6 mice/group)

Minor comments:

1. *Figure 1A, please enhance the magnification and show the IL11 expression pattern on the plasma membrane of hepatocytes and IL6R on Kupffer cells.*

Author response: We localised IL11RA to hepatocytes using immunohistochemistry, immunofluorescence and FACS. We have not studied IL6R in Kupffer, or any other immune-type, cells in this study.

2. *Figure 1A, the size of the scale bar is missing.*

Author response: We have added the scale bar size information for images shown in **Fig. 1A**.

3. *The statement that “only a few hepatocytes expressed low amounts of IL6R” is not entirely correct. In Figure 1B, most of the cells stained by IL6R still have higher fluorescence intensity than the IgG control. This means, most of the cells are still expressing IL6R but at a much lower level than IL11RA and gp130.*

Author response: We thank the Reviewer for pointing this out. In the drafting of the original manuscript there was mis-transcription between FACS studies of cell death (% PI positive) and the amount of Alexa Fluor 488 detected (fluorescence intensity / cell) in cells stained for IL11RA or IL6R. Reporting % positivity for cells stained for membrane receptors was incorrect and we apologize for this oversight. We have amended the figures to show histograms of fluorescence intensity for IL6R, IL11RA, and gp130 in both hepatocytes and THP-1 (**Fig. 1B; Supplementary Fig. 3A**). We have also amended the text accordingly. We thank the Reviewer again for noticing this error.

Figure 1B

Supplementary Figure 3A

4. *Although it is in medium, the true value of ALT should be indicated in Figure 1 as in Figure 2.*

Author response: We now show ALT values (ALT concentration (U/L)) throughout the revised manuscript.

5. Palmitate induces less than 1ng/ml of IL11. Based on the data in Figure 1 and S2, IL11 at this concentration did not induce a major difference in ALT (Figure S2B). Therefore, it is difficult to conclude that IL11-IL11RA signaling is a major contributor to palmitate-induced cell death.

Author response: There are differences between exogenous recombinant IL11 added to the media and endogenous IL11 acting in an autocrine loop on membrane receptors and we believe it is not meaningful to compare levels. The proof that IL11, at the levels measured in the media, has effect is apparent by the use of antibodies that block IL11RA (**Fig. 2**) or IL11 itself (**Rebuttal Fig. 2.2**), which inhibit IL11-induced cell death in palmitate-loaded hepatocytes.

Rebuttal Fig. 2.2 Effect of IL11 inhibition by X203 (IL11 antibody) on palmitate-induced ALT release (n=3).

6. Figure 2H, it is better to show reduced and total glutathione.

Author response: We have now performed reduced GSH assay and added the data as part of **Fig. 2H** as shown below.

7. Figure 2I, DCFDA should be stained together with DAPI to show the same cell density.

Author response: We have added the bright field images which show similar cell density for the DCFDA data shown in **Fig. 2I** as **Supplementary Fig. 5B**.

8. Figure 2K, increase the magnification of the Oil Red O staining as it is difficult to assess the changes in fatty acid uptake. It is also recommended to measure intracellular TG and NEFA (maybe normalized by protein concentration) to strengthen the conclusion.

Author response: At the Reviewer's request we now provide a higher magnification of Oil Red O staining (ORO) images while still showing a representative field of view (**Supplementary Fig. 5E**). We agree with the Reviewer's sentiment that while ORO staining is useful to show lipid accumulation qualitatively, quantitative assessment of TGs should also be provided. We now show new data on intracellular triglyceride (TG) levels in **Supplementary Fig. 5F**. This reveals that the levels of intracellular TG in X209-treated palmitate loaded hepatocytes are lower than controls ($P=0.0003$). This is an important finding and reflects improved mitochondrial function and beta-oxidation of fatty acids that is associated with inhibition of IL11 signaling in lipotoxic hepatocytes (**Fig 2K; Supplementary Fig. 5G-H**). New data on these various aspects (along with serum beta-hydroxybutyrate levels in *in vivo* studies (**Supplementary Fig. 9H**) is now presented and discussed in the revised manuscript and shown in part below.

9. *It is not clear why FASN is mentioned in the results. This is a protein involved in lipogenesis. However, it is not commented anywhere else.*

Author response: We have now removed any mention of FASN from the revised manuscript.

10. *Figure 3, again, there is no evidence showing that sIL11RA is present in the environment.*

Author response: We refer the reviewer to our response to question 1 and also the fact that we now show that sIL11RA levels are increased in the serum of mice fed with WDF (**Rebuttal Figure 2.2**) and that hepatocytes produce sIL11RA themselves (**Rebuttal Figure 2.1**).

11. *Figure 4C, absolute b.w. number is recommended, and the NC WT curve is missing.*

Author response: We have added a panel showing absolute body weight (g) as **Supplementary Fig. 8B**. NC WT curve (dotted grey) was behind and overlapped with NT CKO curve in the previous version of the figure. For clearer visualization, the NC WT line (dotted grey) has now been brought forward.

Supplementary Figure 8B

12. Figure 4D, not clear the H&E is from which time point.

Author response: H&E staining was done on livers from mice after 4 weeks of HFCMD. This has now been made clear in **Fig. 4D**.

Reviewer #3 (Remarks to the Author):

The study by Dong et al. aimed to examine a role for IL11 signaling in hepatocyte lipotoxicity in relation to the pathogenesis of non-alcoholic steatohepatitis (NASH). For this purpose, the authors performed animal studies involving various NASH models and showed that hepatocyte-specific deletion of IL11ra1 protects mice from all aspects of NASH. The authors also showed that restoration of IL11 cis-signaling in hepatocytes only in mice globally deleted for Il11ra1 reconstitutes steatosis and inflammation. Based on their results, the authors conclude that autocrine IL11-mediated cell death underlies hepatocyte lipotoxicity and that liver fibrosis and inflammation occur subsequently. Overall, this is an interesting study. However, the study overly stated the role of autocrine IL11 activity in hepatocytes in promoting steatohepatitis and nearly ignored completely the roles of factors derived from IL11-driven hepato-lipotoxicity on activating liver Kupffer cells/macrophages and hepatic stellate cells.

Specific comments

1. *This manuscript has included strong data to support a detrimental role for IL11 cis-signaling in hepatocytes in promoting NASH.*

Author response: We thank the Reviewer for this comment.

2. *Since IL11 cis-signaling is critical to hepatocyte lipotoxicity, it is important to show whether IL11RA expression in liver sections from NASH patients differs from that in normal liver sections.*

Author response: We have previously shown that IL11 is elevated in human NASH⁵ and point out that upregulation of the cytokine itself would be sufficient to stimulate IL11 *trans* signaling, by forming more IL11:IL11RA complexes, potentially in the absence of changes in levels of IL11RA itself. This said, we have now performed immunohistochemistry (IHC) staining of IL11RA on liver sections from healthy individuals and patients with NASH and found that IL11RA expression is increased in human NASH. In addition we also observed a similar effect in mice with NASH following WDF feeding. These data have been added as **Supplementary Fig. 2B and C** in the revised manuscript.

Supplementary Figure 2B-C

3. Data in Fig. 1L, K and Fig. 2G,H,J indicate a predominant role for IL11 cis-signaling in causing hepatocyte death.

Author response: We agree and believe there is no conclusive data to support a role for IL11 trans-signaling in hepatocyte lipotoxicity (or NASH).

4. Data in Fig. 3B is quite interesting. However, this reviewer is curious if IL11RA is differentially expressed in liver sections from WDF-Null mice vs. that from WDF-sgp130 mice.

Author response: We have performed additional experiments to address this point. IL11RA is elevated in murine NASH and its levels are unaffected by sgp130 expression (**Rebuttal Fig. 3.1**).

Rebuttal Figure 3.1. Representative images showing IHC staining of IL11RA1 in livers of mice fed with normal chow diet (NC mice) or WDF in the presence or absence of sgp130 expression.

It is also important to demonstrate the expression pattern of IL11RA in hepatocytes and non-parenchymal cells (NPCs) from liver sections; given the particular importance of NPCs in promoting NASH in response to hepatocyte-derived factors.

Author response: At higher power we now show that the IL11RA is largely localised to hepatocytes in the human liver (**revised Fig. 1A**). This is consistent with the staining from the Human Protein Atlas, as shown below (Rebuttal Fig. 3.2). It is the case that IL11RA is expressed on other liver cells - notably hepatic stellate cells - but by gross histochemistry this is not apparent but something we refer to and reference.

Rebuttal Figure 3.2. Image of IL11RA expression that is mostly seen in hepatocytes in the human liver. <https://www.proteinatlas.org/ENSG00000137070-IL11RA/tissue/liver#img>

In the revised manuscript we now state:

“In both human and mouse liver sections there was limited staining of IL6R but robust expression of IL11RA, which appeared mostly localised to hepatocytes (**Fig. 1A**; **Supplementary Fig. 2A**), which is consistent with IHC staining data from the human protein atlas using two additional antibodies (CAB032830 and HPA036652; <https://www.proteinatlas.org>).”

5. While data in Fig. 3O,P indicated increased liver inflammation in either WDF-Null mice or WDF-sgp130 mice vs. NC-Null mice, it is not clear about the proportional contribution of increased inflammatory responses in hepatocytes vs. NPCs to liver inflammation.

Author response: The Reviewer raises an important point in that inflammation in the liver is regulated by a large and diverse set of cells including hepatocytes, hepatic stellate cells, Kupffer cells, cholangiocytes, resident and circulating innate immune cells, resident and circulating adaptive immune cells, among others. We do not attempt to address the variable contributions of all these cell types to the phenotypes we report in the current manuscript. However, we point to **Fig. 4J, 5L** and **Supplementary Fig. 8C and 9C** where we show that specific deletion of *Il11ra* in hepatocytes reduces inflammation in the steatotic liver. Thus, we can conclude that

IL11 signaling in hepatocytes (the focus of our study) plays a role for liver inflammation. We agree that the contribution of other cells will also be of relevance, although not studied here. We now discuss these matters more fully in the revised discussion.

6. Fig. 4C,D: HFMCD-fed CKO mice displayed significantly decreased hepatic steatosis while showing no decrease in body weight. In contrast, HFMCD-WT mice displayed massive hepatic steatosis while also showing a marked decrease in body weight. This reviewer is curious about the mechanisms by which hepatocyte-specific IL11RA disruption decreases hepatic steatosis. This reviewer is also curious if increased adipose tissue lipolysis and fat flow to the liver in WT mice contributed to hepatic steatosis and whether this mechanism was impaired in HFMCD-CKO mice.

Author response: The Reviewer raises points that we are also very interested to understand but will take some years to decipher fully. While we did not set out to study the effects of *Il11ra* deletion in hepatocytes on weight in mice on the HFMCD diet, we were surprised by the data and felt it was important to include them in a main figure. If the Reviewer feels that the weight of the data are distracting from the main findings, we could remove them. As to how the CKO mice are protected from weight loss, we do not know about the mechanism. HFMCD is a model of cytokine-associated cachexia and it is possible that the lesser inflammation is linked to weight gain. Having said that, we have unpublished data showing the same effect with IL11 and IL11RA antibody therapy in the HFMCD model.

During this revision, we have performed a number of new studies showing that inhibition of IL11 signaling in lipotoxic hepatocytes is associated with improved mitochondrial oxidative function, increased fatty acid oxidation and reduced intracellular triglyceride accumulation (**Fig.2K, Supplementary Fig. 5F-H**). Thus, we surmise that IL11-dependent effects impair the ability of hepatocytes to adapt to lipid loading. As these data were generated *in vitro*, this excludes fat flow into / out of the liver as a potential confounding factor. These new data are presented and discussed in the revised manuscript.

Figure 2K

Supplementary Figure 5F

Supplementary Figure 5G-H

7. For mouse models in Fig. 4, what were the serum levels of IL11?

Author response: These data have now been generated and have been added to the revised manuscript as **Supplementary Fig. 8A and 9A**.

Supplementary Figure 8A

Supplementary Figure 9A

8. For data in Fig. 4 and Fig 5: it appears that upon HFMCD feeding CKO mice recovered from weight loss or gained body weight compared with WT mice. However, upon WDF feeding, CKO mice revealed a smaller gain in body weight compared with WT mice. Are there any explanations for why CKO mice responded differently to different diets? This is important because adiposity or fat mass is a key factor determining hepatic steatosis.

Author response: We offer our thoughts on the differential effects. In the HFMCD diet, wild-type mice develop severe and acute steatohepatitis and lose up to 40-50% of their body weight within a matter of weeks. CKO mice on HFMCD, which do not suffer from steatohepatitis, are systemically well and able to maintain appetite and weight on this toxic diet. It is indeed the case that inhibition of IL11 signaling in this model is associated with greater food intake (unpublished).

The WDF model is more akin to human NASH. Mice on WDF develop insidious weight gain, steatosis, metabolic derangement and mild hepatitis. In this model, loss of IL11 signaling in hepatocytes is associated with preserved liver metabolism, improved serum metabolic profiles and elevated serum beta-hydroxybutyrate (BHB). While we cannot adequately address weight loss on the WDF diet experimentally in the current manuscript we believe that improved hepatocyte mitochondrial function and fatty acids oxidation plays an important role and mention this in the revised discussion.

9. For Fig. 5E: what are mechanisms for decreased hepatic steatosis in WDF-CKO mice?

Author response: We refer the Reviewer to our response to question 6 above where we mention: “ During this revision, we have performed a number of new studies showing that inhibition of IL11 signaling in lipotoxic hepatocytes is associated with improved mitochondrial oxidative function, increased fatty acid oxidation and reduced intracellular triglyceride accumulation (**Fig.2K, Supplementary Fig. 5F-H**). Thus, we surmise that IL11-dependent effects impair the ability of hepatocytes to adapt to lipid loading.”

Furthermore, while we have now removed data on fatty acid synthetase (FASN) from the revised manuscript, following a comment from Reviewer 2, it is apparent that *de novo* lipogenesis plays a role. FASN expression, which is increased in steatotic livers and contributes to *de novo* lipogenesis¹⁰, is decreased in lipid-loaded hepatocytes and in NASH following inhibition of IL11 signaling (**Rebuttal Fig. 3.3**). Hence, inhibition of IL11 in the context of lipotoxicity both increases fatty acid oxidation and also diminishes *de novo* lipogenesis in hepatocytes.

Rebuttal Figure. 3.3. Western blots showing expression of FASN and GAPDH from (A) livers of WT and CKO mice on NC and WDF and from (B) palmitate-treated primary human hepatocytes in the presence of IgG, X209, or sgp130.

10. Fig. 6: the data are strong in terms of validating a role for activation of IL11 cis-signaling in promoting NASH.

Author response: We thank the Reviewer for recognizing this.

11. Fig. 7: the paracrine actions of hepatocytes on HSCs have not been validated. Indeed, hepatocyte factors, generated in response to activation of IL11 cis-signaling, could act on HSCs and accounts for, in large part, the fibrogenic activation of HSCs.

Author response: In the original manuscript, we showed that prevention of IL11 cis-signaling in hepatocytes reduced fibrosis suggestive of paracrine activity of hepatocyte-secreted IL11 on HSCs. As our studies were primarily focused on hepatocyte-specific effects we did not originally extend studies to HSCs. Based on the Reviewer's suggestion, we performed new experiments to examine whether fibrosis is dependent on IL11 signaling *per se* or if other hepatocyte-derived factors are involved. We cultured HSCs with conditioned media from either control or palmitate-treated hepatocytes and found that media from lipotoxic hepatocytes strongly induced ACTA2 and Collagen expression in HSCs (**Fig. 2L; Supplementary Fig.5I and J**). Addition of X209 to the conditioned media blocked ACTA2 and Collagen, whereas sgp130 had no effect. These data demonstrate that lipotoxic hepatocytes release IL11 that acts in a paracrine manner to activate IL11 cis-signaling in HSCs, consistent with our earlier *in vivo* data.

Figure 2L

Supplementary Figure 5I-J

12. *Similar to that described in Point 11, the paracrine actions of hepatocyte-derived factors on liver Kupffer cells/macrophages could account for, in large part, liver inflammation. This point has not even been mentioned.*

Author response: We refer to point 5 above. The major objective of the current study was to examine the effects of IL11 *cis* and *trans* signaling in hepatocytes on lipotoxicity and NASH. We show that hepatocytes themselves play an important role in the regulation of some inflammatory factors (e.g. TNF α , CCL2 and CCL5). While we now partially extend our studies to HSCs, as suggested by the Reviewer, we do not propose to examine IL11 effects in the various immune cells in the liver, which we believe is beyond the scope of the current study. We have discussed this matter in the limitations (discussion) section of the manuscript to ensure that this important aspect is not overlooked.

References

1. Schafer, S. *et al.* IL-11 is a crucial determinant of cardiovascular fibrosis. *Nature* **552**, 110–115 (2017).
2. Cook, S. A. & Schafer, S. Hiding in Plain Sight: Interleukin-11 Emerges as a Master Regulator of Fibrosis, Tissue Integrity, and Stromal Inflammation. *Annu. Rev. Med.* **71**, 263–276 (2020).
3. Widjaja, A. A. *et al.* Redefining Interleukin 11 as a regeneration-limiting hepatotoxin. doi:10.1101/830018.
4. Nishina, T. *et al.* Interleukin-11 links oxidative stress and compensatory proliferation. *Sci. Signal.* **5**, ra5 (2012).
5. Widjaja, A. A. *et al.* Inhibiting Interleukin 11 Signaling Reduces Hepatocyte Death and Liver Fibrosis, Inflammation, and Steatosis in Mouse Models of Non-Alcoholic Steatohepatitis. *Gastroenterology* (2019) doi:10.1053/j.gastro.2019.05.002.
6. Ng, B. *et al.* Interleukin-11 is a therapeutic target in idiopathic pulmonary fibrosis. *Sci. Transl. Med.* **11**, (2019).
7. Newberry, E. P. *et al.* Hepatocyte and stellate cell deletion of liver fatty acid binding protein reveals distinct roles in fibrogenic injury. *FASEB J.* **33**, 4610–4625 (2019).
8. Lokau, J. *et al.* Proteolytic Cleavage Governs Interleukin-11 Trans-signaling. *Cell Rep.* **14**, 1761–1773 (2016).
9. Kammoun, H. L. *et al.* Over-expressing the soluble gp130-Fc does not ameliorate methionine and choline deficient diet-induced non alcoholic steatohepatitis in mice. *PLoS One* **12**, e0179099 (2017).
10. Dorn, C. *et al.* Expression of fatty acid synthase in nonalcoholic fatty liver disease. *Int. J. Clin. Exp. Pathol.* **3**, 505–514 (2010).

Reviewers' Comments:

Reviewer #1:

Remarks to the Author:

The authors have responded to all comments raised by the reviewer, thus, the manuscript will be suitable for publication in Nature Communications.

Reviewer #2:

Remarks to the Author:

None

Reviewer #3:

Remarks to the Author:

This reviewer is satisfied by the revision. There is no further comment.

Point-by-point responses to the comments made by Reviewers at Nature Communications

Reviewer #1 (Remarks to the Author):

The authors have responded to all comments raised by the reviewer, thus, the manuscript will be suitable for publication in Nature Communications.

Author response: We thank the Reviewer for the recommendation.

Reviewer #2 (Remarks to the Author):

None

Author response: We thank the Reviewer for this comment.

Reviewer #3 (Remarks to the Author):

This reviewer is satisfied by the revision. There is no further comment.

Author response: We thank the Reviewer for this comment.